# Aortic disease in Marfan syndrome is caused by overactivation of sGC-PRKG signaling by NO

Andrea de la Fuente-Alonso[1,2,11], Marta Toral [1,2,11], Alvaro Alfayate[1,2,3], María Jesús Ruiz-Rodríguez [1,2], Elena Bonzón-Kulichenko [2,3], Gisela Teixido-Tura [2,4], Sara Martínez-Martínez[1,2], María José Méndez-Olivares[1,2], Dolores López-Maderuelo[1,2], Ileana González-Valdés[3], Eusebio Garcia-Izquierdo[5], Susana Mingo[5], Carlos E. Martín[6], Laura Muiño-Mosquera[7], Julie De Backer [7], J. Francisco Nistal [2,8], Alberto Forteza[6], Arturo Evangelista[2,4], Jesús Vázquez [2,3], Miguel R. Campanero [2,9,10,12 ✉] & Juan Miguel Redondo [1,2,12✉]

Thoracic aortic aneurysm, as occurs in Marfan syndrome, is generally asymptomatic until dissection or rupture, requiring surgical intervention as the only available treatment. Here, we show that nitric oxide (NO) signaling dysregulates actin cytoskeleton dynamics in Marfan Syndrome smooth muscle cells and that NO-donors induce Marfan-like aortopathy in wild-type mice, indicating that a marked increase in NO suffices to induce aortopathy. Levels of nitrated proteins are higher in plasma from Marfan patients and mice and in aortic tissue from Marfan mice than in control samples, indicating elevated circulating and tissue NO. Soluble guanylate cyclase and cGMP-dependent protein kinase are both activated in Marfan patients and mice and in wild-type mice treated with NO-donors, as shown by increased plasma cGMP and pVASP-S239 staining in aortic tissue. Marfan aortopathy in mice is reverted by pharmacological inhibition of soluble guanylate cyclase and cGMP-dependent protein kinase and lentiviral-mediated *Prkg1* silencing. These findings identify potential biomarkers for monitoring Marfan Syndrome in patients and urge evaluation of cGMP-dependent protein kinase and soluble guanylate cyclase as therapeutic targets.

[1] Gene regulation in cardiovascular remodeling and inflammation group, Centro Nacional de Investigaciones Cardiovasculares (CNIC), Madrid, Spain. [2] Centro de Investigación Biomédica en Red de Enfermedades Cardiovasculares (CIBERCV), Madrid, Spain. [3] Cardiovascular Proteomics Laboratoy, CNIC, Madrid, Spain. [4] Servei de Cardiologia, Hospital Vall d'Hebron, Barcelona, Spain. [5] Cardiology Department, Hospital Universitario Puerta de Hierro, Madrid, Spain. [6] Cardiac Surgery Department, Hospital Universitario Puerta de Hierro, Madrid, Spain. [7] Center for Medical Genetics Ghent, Ghent University Hospital, Ghent, Belgium. [8] Cardiovascular Surgery and Department of Physiology and Pharmacology, Hospital Universitario Marqués de Valdecilla, IDIVAL, Facultad de Medicina, Universidad de Cantabria, Santander, Spain. [9] Department of Cancer Biology, Instituto de Investigaciones Biomedicas Alberto Sols, Consejo Superior de Investigaciones Científicas–Universidad Autónoma de Madrid, Madrid, Spain. [10] Present address: Centro de Biología Molecular Severo Ochoa, Consejo Superior de Investigaciones Científicas–Universidad Autónoma de Madrid, Madrid, Spain. [11] These authors contributed equally: Andrea de la Fuente-Alonso, Marta Toral. [12] These authors jointly supervised this work: Miguel R. Campanero, Juan Miguel Redondo. ✉email: mcampanero@cbm.csic.es; jmredondo@cnic.es

Thoracic aortic aneurysm and dissection (TAAD) is a major cause of morbidity and mortality in developed countries[1]. TAAD is characterized by progressive vessel dilation associated with vascular smooth muscle cell (VSMC) dysfunction and destructive extracellular matrix remodeling[2,3]. TAAD is often asymptomatic until the aorta dissects or ruptures. Approximately one in four cases of TAAD have a known genetic basis and can be categorized as either syndromic (showing prominent phenotypic features of a systemic connective tissue disorder such as Marfan syndrome [MFS]) or nonsyndromic, with an essentially vascular phenotype[3]. TAAD and rupture account for >90% of MFS patient deaths[4,5]. MFS is an inherited autosomal dominant disease caused by pathogenic variants of the fibrillin-1-encoding gene FBN1. Fibrillin-1 is a major component of extracellular microfibrils, providing a scaffold for elastic-fiber formation and maturation[6]. Mutant fibrillin-1 disrupts microfibril formation, leading to medial degeneration, which destabilizes the aortic wall, rendering the aorta vulnerable to hemodynamic injury[7,8]. Medial degeneration is characterized by poor alignment of elastin filaments, disorganization of lamellar units, VSMC death, and proteoglycan accumulation[9]. Proteoglycans play essential roles in the preservation of aortic structure and function by regulating elastic fiber assembly and smooth muscle cell proliferation[10–12].

Current therapy for the cardiovascular complications of MFS consists of strict follow-up, life-style advice, medical treatment to slow the rate of aortic root dilation, and surgery to prevent dissection or rupture[13]. Pharmacological therapy is usually based on ß-adrenoceptor blockade[14]. However, this treatment neither halts abnormal aortic growth nor prevents aortic dissection or death[14]. Moreover, some studies suggest that ß-adrenergic blockers do not even slow the rate of aortic growth in MFS[15,16]. There is therefore a need for effective pharmacological strategies to manage TAAD in MFS patients.

MFS is accompanied by impaired aortic contractility[17], caused by as yet unknown molecular mechanisms. In small arteries, VSMC relaxation and vasodilatation are induced by nitric oxide (NO), produced in endothelial cells by constitutively expressed endothelial NO synthase (eNOS; also called NOS3). NO can also be produced by constitutively expressed NOS of neuronal origin (nNOS; also called NOS1) or by inducible NOS (iNOS; also called NOS2)[18]. NOS2 expression is induced in MFS patients and in a mouse model of MFS, and TAAD is reversed in the mouse model by pharmacological NOS2 inhibitors, raising the possibility that blocking NOS2 activity could be a promising treatment for TAAD[19]. However, the mechanisms by which NOS2 contributes to TAAD in MFS remain unclear. Increased NOS2-derived NO levels stimulate soluble guanylate cyclase (sGC) to generate cGMP, which in turn activates cGMP-dependent protein kinase G (PRKG), which modulates the contractility of resistance vessels by regulating actin filament and myosin dynamics[20]. NO also regulates numerous physiological processes in a cGMP-independent manner, including mechanisms based on S-nitrosylation[21,22] and, in the presence or excess reactive oxygen species (ROS), protein tyrosine nitration[23].

In response to limited availability of the substrate L-arginine or the cofactor tetrahydrobiopterin (BH4)[24–26], or high ROS levels[27], all NOS enzymes generate superoxide anion instead of NO, resulting in oxidative stress. As ROS levels are elevated in the MFS aorta[17,28], NOS2 might be responsible for superoxide anion generation in this scenario. Moreover, ROS activate PRKG by direct oxidation[29], raising questions about whether TAAD generation in MFS is mediated by NOS2-derived superoxide or NO and whether cGMP or PRKG act as regulators in this process.

Here, we show that the NO–sGC–PRKG signaling pathway is activated in MFS mice and MFS patients, and demonstrate that this pathway mediates aortopathy in a mouse model of MFS.

## Results

**NO signaling pathway activation dysregulates actin cytoskeleton dynamics in MFS VSMCs.** To investigate whether the NO–sGC–PRKG signaling pathway is activated in MFS, we measured VASP-S239 phosphorylation (pVASP-S239) as a readout of PRKG activity[30] in primary aortic VSMCs from wild-type (WT) mice[31,32] and a mouse model of MFS (Fbn1$^{C1039G/+}$ mice)[33] cultured in the presence of inhibitors of NOS (L-NAME), sGC (ODQ), and PRKG (KT5823) (Fig. 1a). As pathway activation controls, we also treated WT cells with the cGMP analog 8-Br-cGMP or the NO-donor DetaNONOate (DetaNO). These agents induced pVASP-S239, and this event was prevented by the corresponding inhibitors ODQ and KT5823 (Fig. 1b, c). Untreated MFS VSMCs had higher pVASP-S239 levels than WT cells (Fig. 1b, c), and these levels were sharply decreased by treatment with ODQ or KT5823 (Fig. 1b, c), strongly suggesting that the NO–sGC–PRKG signaling pathway is activated in MFS VSMCs.

VASP-S239 phosphorylation impairs actin filament formation[34,35], and we therefore investigated whether pVASP-S239 regulation by the NO–sGC–PRKG signaling pathway negatively correlated with actin filament formation. Phalloidin staining showed a sharp decrease of filamentous actin (F-actin) accumulation in WT VSMCs treated with DetaNO or 8-Br-cGMP, correlating with low F-actin in untreated MFS cells (Fig. 2a, b). Moreover, PRKG inhibition by KT5823 in MFS VSMCs markedly increased F-actin formation to levels similar to those found in control cells (Fig. 2b), supporting the notion that the NO signaling pathway is critical for actin cytoskeleton dynamics regulation in MFS.

Regulation of actin cytoskeleton dynamics is critical for effective cell contraction; moreover, the expression of contractile protein markers, including alpha-smooth muscle actin (Acta2), smooth muscle protein 22 alpha (Tagln2), and calponin-1 (Cnn1), is increased in the aorta of MFS mice relative to WT mice[36]. Given these observations, we assessed the role of the NO–sGC–PRKG pathway in regulating the expression of these markers. RT-qPCR analysis showed a substantial increase in Acta2, Cnn1, and Tagln2 mRNA in WT VSMCs treated with DetaNO or 8-Br-cGMP (Fig. 2c) and also in MFS VSMCs (Fig. 2d). Consistent with a role for the NO pathway in regulating the expression of these markers, pharmacological PRKG inhibition decreased their expression to normal levels (Fig. 2d). These results indicate that some of the enhanced SMC markers characteristic of MFS VSMCs can be reverted by NO–sGC–PRKG targeting.

MFS smooth muscle cells harboring the FBN1$^{C1242Y}$ pathogenic variant show defective deposition of fibrillin-1[37]. However, we found no notable effect on extracellular fibrillin-1 deposition in WT VSMCs treated with with 8-Br-cGMP (Supplementary Fig. 1).

Together these results strongly suggest that overactivation of the NO–sGC–PRKG pathway in MFS VSMCs switches their phenotype by dysregulating the contractile machinery, including actin cytoskeleton dynamics, without affecting microfibril formation.

**Supraphysiological NO levels induce an MFS-like aortic pathology.** VSMCs from MFS patients and the Fbn1$^{C1039G/+}$ mouse model (MFS mice) express high levels of NOS2[19], a NOS enzyme that produces large amounts of NO[38]. We therefore investigated whether supraphysiological NO levels affect aortic homeostasis in healthy mice. As an NO source, we used isosorbide mononitrate (ISMN), a long-lasting nitrate that is metabolized to NO within VSMCs and induces dilation of systemic

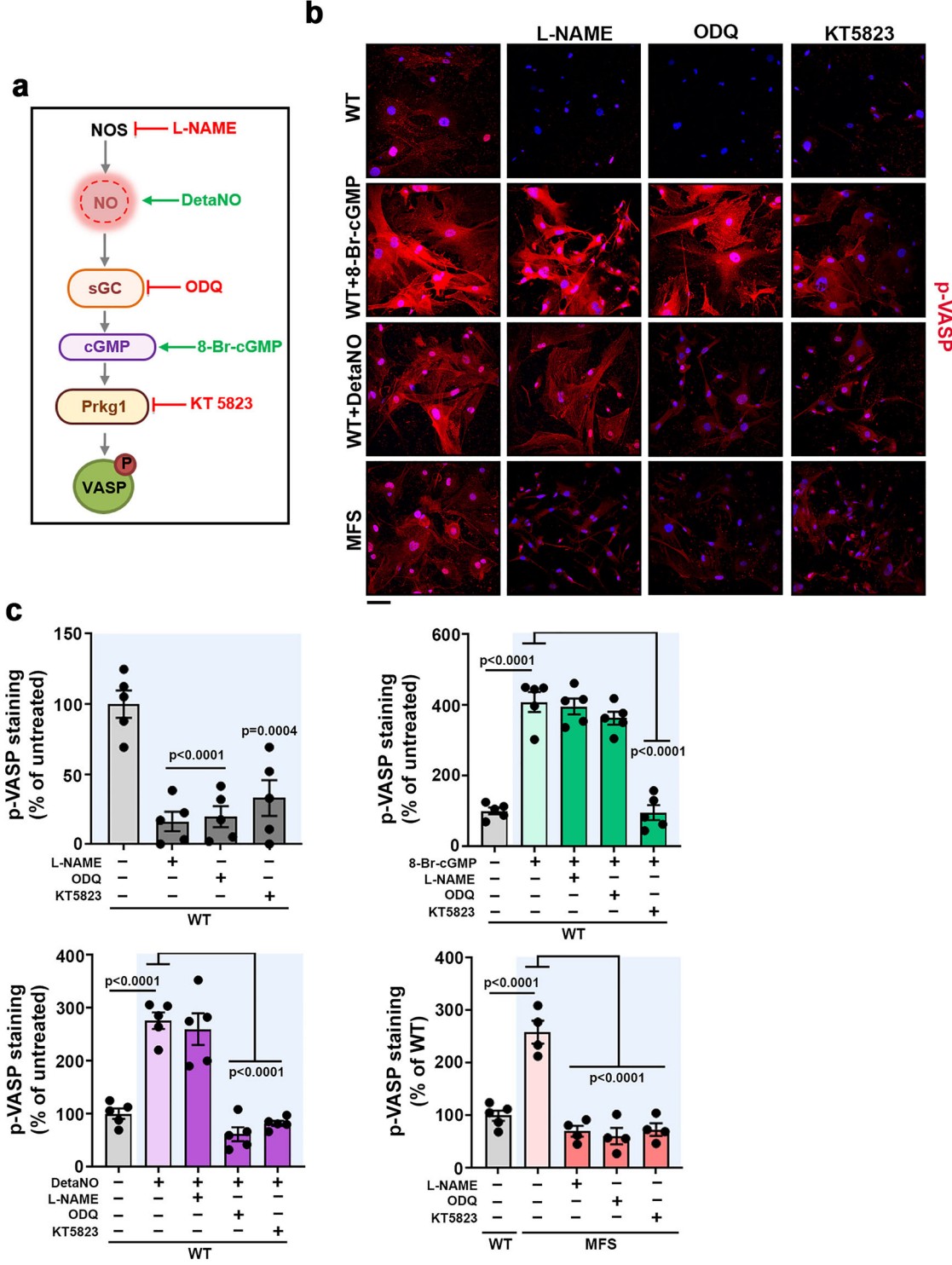

**Fig. 1 Pharmacological inhibition of signaling components of the NO–sGC–PRKG pathway decreases pVASP-S239 induction in VSMCs from MFS mice.**
**a** NO signaling components and the targets of pharmacological stimuli (green) and inhibitors (red). **b** Representative images of pVASP-S239 immunofluorescence (red) and DAPI-stained nuclei (blue). **c** Quantification of pVASP-S239 immunofluorescence in WT and MFS VSMCs treated with 300 μM L-NAME, 10 μM ODQ, or 1 μM KT5823 for 1 h before stimulation for 5 minutes with 100 μM 8-Br-cGMP or 100 μM DetaNO, as indicated. **b, c** Five independent cell batches were used for all conditions except for MFS VSMCs ($n = 4$). Scale bar, 50 μm. Data are shown relative to untreated WT cells as mean ± s.e.m. Each data point denotes the mean value from an independent experiment. Differences were analyzed by one-way ANOVA with Dunnett's post-hoc test ($p$-values are shown). Source data are provided in the Source Data file.

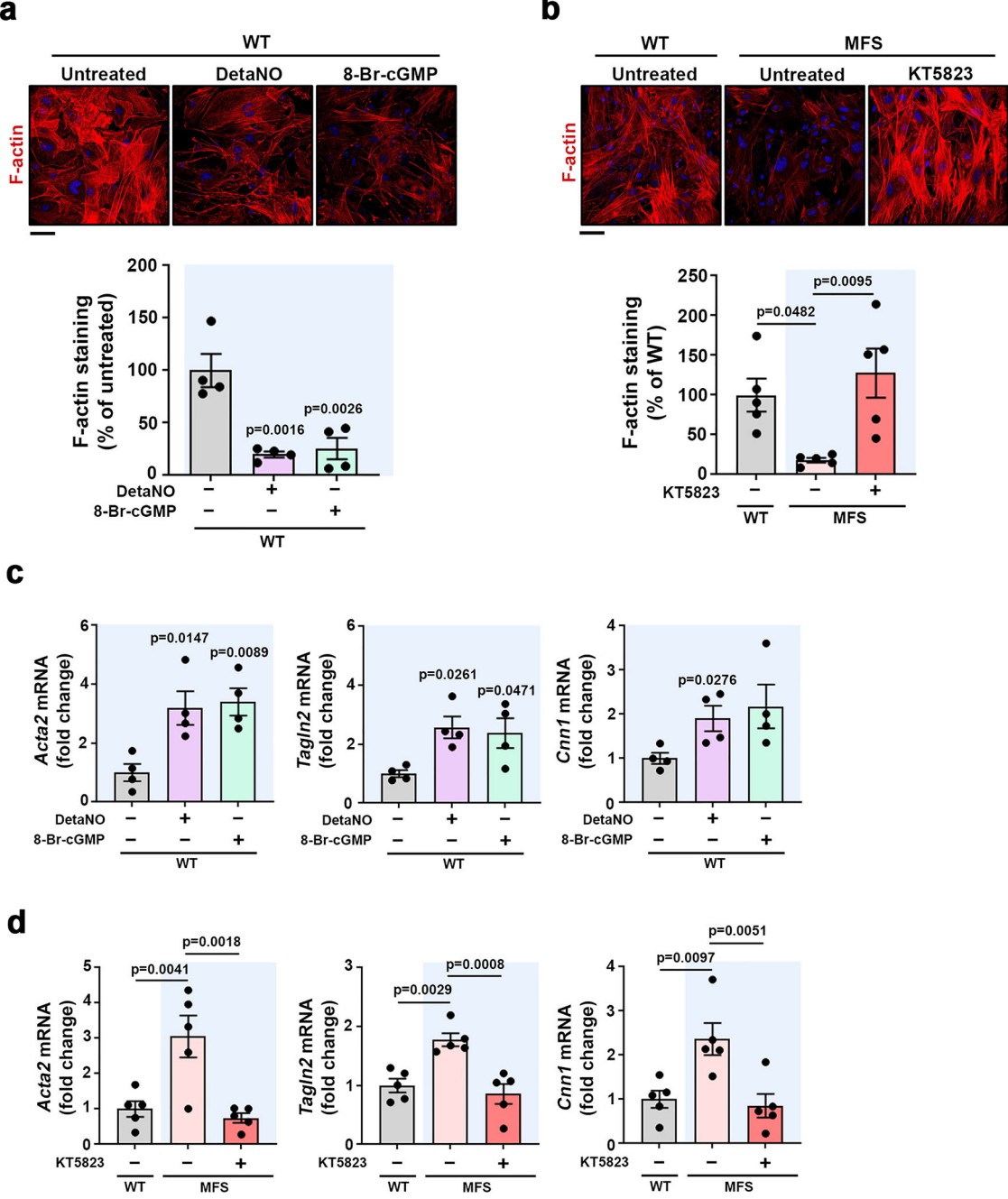

**Fig. 2 The NO–sGC–PRKG pathway modulates the contractility phenotype of VSMCs. a, b** Representative images of F-actin staining (red) and DAPI-stained nuclei (blue) and F-actin quantification in **a** WT VSMCs treated as indicated for 5 min ($n = 4$ independent cell batches per condition) and **b** WT and MFS VSMCs untreated or treated with KT5823 for 24 h ($n = 5$ independent cell batches per condition). Scale bar, 50 µm. Data are shown relative to untreated WT cells as mean ± s.e.m. Each data point denotes the mean value from an independent experiment. Differences were analyzed by one-way ANOVA with Tukey's post-hoc test (p-values are shown). **c, d** RT-qPCR analysis of *Acta2, Cnn1,* and *Tagln2* mRNA expression in **c** WT VSMCs treated as indicated for 4 h and **d** WT or MFS VSMCs treated as indicated for 24 h ($n = 4$ independent cell batches per group in **c** and $n = 5$ independent cell batches per group in **d**). mRNA amounts were normalized to *Gapdh* expression (mean ± s.e.m.). Each data point denotes the mean value from an independent experiment. Differences were analyzed by one-way ANOVA followed by Dunnett's post-hoc test (p-values are shown). Source data are provided in the Source Data file.

and coronary vascular beds[39]. We performed in vivo longitudinal dose-response studies to characterize the effects of infusing ISMN for 7 days on the aortic phenotype of WT C57BL/6 mice (Fig. 3a). Since there were no previous reports on the effect of ISMN on aortic phenotype, and the normal clinical dose of ISMN used to treat chronic stable angina pectoris ranges from 0.5 to 4 mg/kg/day, we tested several doses from 1 to 50 mg/kg/day. ISMN induced a dose-dependent decrease in systolic blood pressure (BP) (Fig. 3b) and dilation of both the ascending aorta (AsAo) and the abdominal aorta (AbAo) (Fig. 3c, d). Aortic diameters at the highest dose were similar to those in MFS mice (Fig. 3e). ISMN doses above 50 mg/kg/day did not induce larger aortic dilations or BP drops (Supplementary Fig. 2), and this dose was therefore used for further experiments.

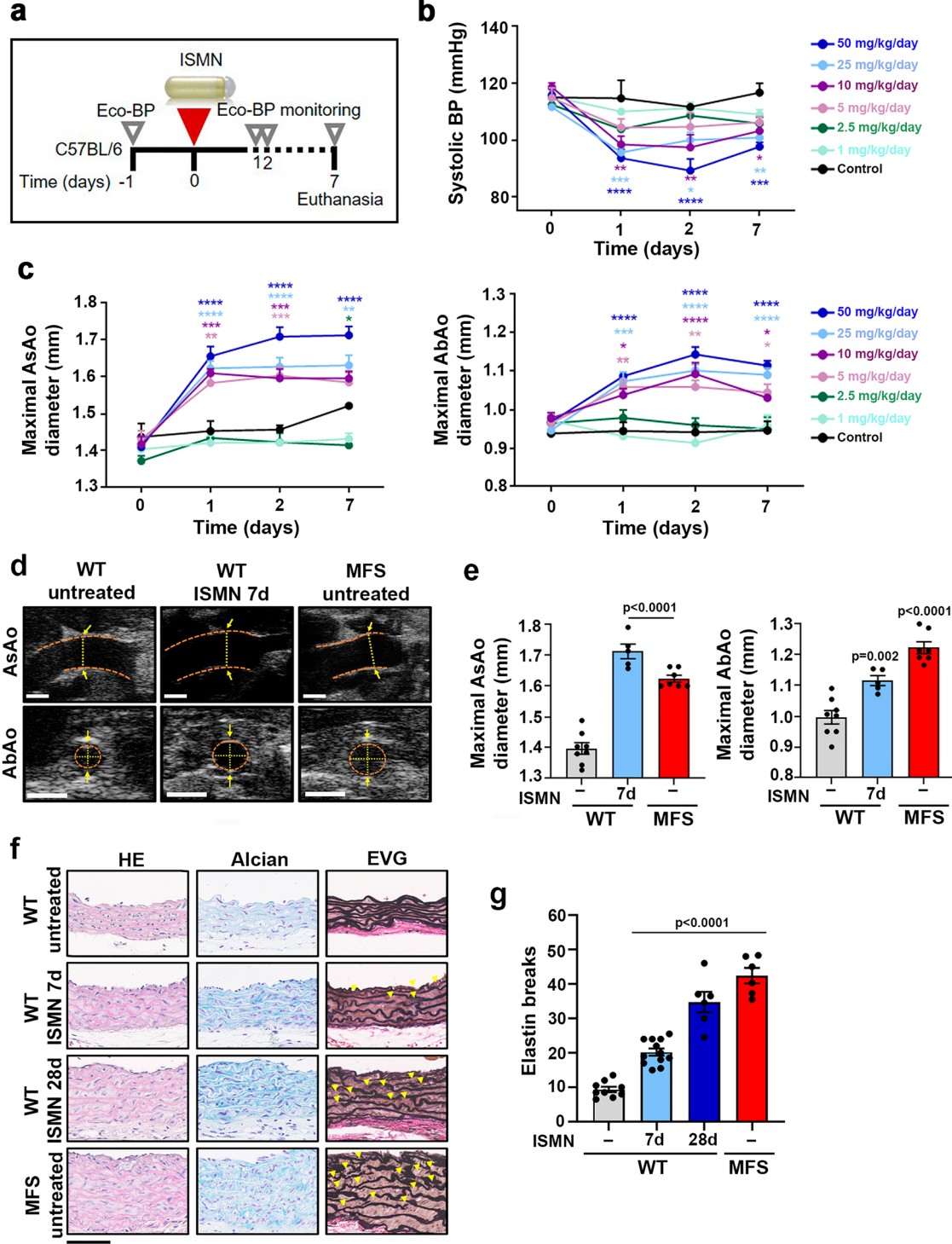

To determine if other NO donors had a similar effect on BP and aortic dilation, we treated WT mice with DetaNO. Unlike ISMN, DetaNO is unstable at 37 °C, and we therefore used osmotic minipumps to infuse it for just 2 days. Like ISMN, DetaNO decreased systolic BP (Supplementary Fig. 3a) and induced marked AsAo and AbAo dilations (Supplementary Fig. 3b-c).

*FBN1* pathogenic variants cause not only aortic dilation, but also medial degeneration characterized by elastic-fiber fragmentation and disarray and proteoglycan accumulation in the aorta, a feature that might predispose to dissection[10]. Histological analysis of the AsAo[40] in ISMN-infused WT mice revealed both proteoglycan accumulation and elastic-fiber fragmentation and disorganization as early as 7 days after treatment initiation, as shown by Alcian Blue and modified Verhoeff elastic-Van Gieson (EVG) staining, respectively (Fig. 3f, g). After 28 days, these features were more pronounced and were comparable to those in age-matched MFS mice (Fig. 3f, g). Similar elastic-fiber alterations were found upon treatment of WT mice with DetaNO (Supplementary Fig. 3d-3e). These results thus indicate that supraphysiological levels of NO are sufficient to induce MFS-like aortopathy in WT mice.

In humans, sustained high plasma nitrate concentrations induce a clinically relevant decay of the vasodilatory effect,

**Fig. 3 ISMN induces an MFS-like aortopathy in WT mice. a** Experimental design. 12-week-old C57BL/6 mice were treated for 7 days (d) with the NO-donor ISMN at 1, 2.5, 5, 10, 25, or 50 mg/kg/day by osmotic minipump infusion. Mice were monitored by Eco-BP (ultrasound and BP analysis) at the indicated times (empty triangles). **b** Systolic BP during ISMN infusion. **c** Maximal AsAo diameter (left) and AbAo diameter (right) during ISMN infusion. Data in **b** and **c** are mean ± s.e.m; $n = 5$ per treated group, $n = 3$ for control group. *$P < 0.05$, **$P < 0.01$, ***$P < 0.001$, and ****$P < 0.0001$ (versus control at each time point) by repeated-measurements two-way ANOVA with Bonferroni's post-hoc test. **d** Representative ultrasound images (orange dashed lines delineate the lumen boundary and yellow dashed lines mark the lumen diameter) and **e** quantification of end-of-experiment maximal AsAo and AbAo diameter in untreated WT mice (−) ($n = 8$), MFS mice ($n = 7$), and WT mice treated with 50 mg/kg/day ISMN for 7 d ($n = 5$). Each data point denotes an individual mouse, whereas histograms show mean ± s.e.m. **f** Representative staining with hematoxylin and eosin (HE) (left), Alcian blue (middle), and elastic van Gieson (EVG) (right) in the AsAo from 9 to 12-week-old untreated WT mice (−), 6 MFS mice, and WT mice treated with 50 mg/kg/day ISMN for 7 d ($n = 12$ mice) or 28 d ($n = 6$ mice). Yellow arrowheads indicate elastin breaks. Scale bar, 50 μm. **g** Quantification of elastin breaks in AsAo sections from 9 to 12-week-old untreated WT mice (−), 6 MFS mice, and WT mice treated with 50 mg/kg/day ISMN for 7 d ($n = 12$ mice) or 28 d ($n = 6$ mice). Each data point denotes an individual mouse, whereas histograms denote mean ± s.e.m. **e, g** Differences were analyzed by one-way ANOVA with Dunnett's post-hoc test (p-values are shown). Source data are provided in the Source Data file.

known as nitrate tolerance[41]. To investigate if continuous ISMN delivery in mice loses efficacy over the long term, we performed a longitudinal study in which ISMN (50 mg/kg/day) was infused for 28 days (Supplementary Fig. 4a). In line with the results shown in Fig. 3, ISMN sharply decreased systolic BP and increased AsAo and AbAo diameters after 1 day of infusion; moreover, these effects were sustained over the remaining days of treatment (Supplementary Fig. 4b-4c), ruling out a tolerance phenomenon.

**The sGC–PRKG signaling pathway is activated in MFS mice and patients**. We next investigated whether the elevated NOS2 expression found in MFS patients and mice results in activation of the sGC–PRKG signaling pathway. sGC generates cGMP, and intracellular cGMP pools are in a dynamic steady-state relationship with plasma cGMP levels[42,43]. We found substantially elevated plasma cGMP in both MFS mice and ISMN-treated WT mice (Fig. 4a), strongly suggesting that sGC is indeed activated in MFS mice. Similar plasma cGMP increases were found upon DetaNO treatment (Fig. 4b). As a readout of PRKG activity, we determined pVASP-S239 levels in aortic tissue. pVASP-S239 levels were very high in the aortas of MFS mice or WT mice treated with ISMN for 7 or 28 days (Fig. 4c, d and Supplementary Fig. 5). The increase in plasma cGMP and aortic pVASP-S239 was not due to increased sGC or Prkg mRNA or protein expression in MFS (Supplementary Fig. 6), suggesting that sGC and Prkg are activated in MFS by post-translational mechanisms.

To determine if the NO–sGC–PRKG pathway is activated in MFS patients, we measured serum or plasma cGMP and PRKG activity in aortic tissue from MFS patients. Plasma or serum cGMP from three independent MFS patient cohorts was markedly higher than in healthy donors (Fig. 5a). PRKG activity was also higher in aortic sections from two MFS patient cohorts than in sections from multiorgan transplant donors. Immunohistochemistry and immunofluorescence revealed higher pVASP-S239 levels in MFS samples (Fig. 5b, c and Supplementary Fig. 7). Together, these data strongly suggest that the sGC–PRKG pathway is activated in MFS patients.

**MFS mice and patients have elevated levels of protein nitration in aortic tissue and plasma**. The lipophilic properties of NO allow it to diffuse from the vessel wall to the vessel lumen[23,44], where it can potentially trigger NO-derived modifications of plasma proteins. These include Cys S-nitrosylation, a labile modification that cannot be directly detected unless a stable derivative is generated[45], and nitration, a stable modification that takes place under pro-oxidant conditions[46]. We carried out a high-throughput proteomics analysis to explore the relative nitration levels of plasma proteins in seven MFS mice and seven WT littermates. As a positive control, we also included 6

DetaNO-treated WT mice. Mass spectrometry analysis of plasma protein digests quantified 43 nitrated peptide species identified in more than 1 MS/MS spectrum from 19 nitrated proteins that included 47 nitrated sites (NS) (Supplementary Data 1 and 2). Of the NS, 88% correspond to nitro-Tyr and 12% to nitro-Trp. The cumulative distribution of nitro-protein quantifications in plasma from NO-donor-treated mice was significantly shifted toward positive values in relation to other plasma proteins (Fig. 6a), indicating that NO treatment produced a generalized increase in nitrated protein abundance. In contrast, the cumulative distribution of all remaining plasma proteins closely followed the expected null-hypothesis distribution (Fig. 6a), demonstrating the accuracy of the model and that the nitration profile was not biased due to the quantitative analysis applied. Interestingly, the relative abundance of nitrated proteins was also elevated in plasma from MFS mice (Fig. 6a), demonstrating an increased nitration in these mice that can be clearly detected in plasma.

To determine if increased aortic NO levels in MFS patients is also indicated by elevated plasma-protein nitration, we performed a quantitative proteomics analysis in plasma samples from 30 healthy donors and 23 MFS patients, identifying 40 nitrated peptide species from 18 nitrated proteins, many of which were also found in the analysis of mouse plasma (Supplementary Data 3 and 4). Among 43 NS identified in these peptides, 66% correspond to nitro-Tyr and 34% to nitro-Trp. Reproducing the results obtained in mice, plasma from MFS patients showed a generalized elevation in the abundance of nitrated proteins relative to controls (Fig. 6b), suggesting that a highly nitrated plasma profile may be a signature of MFS and strongly supporting the notion that NO levels are increased in MFS. To provide a set of candidate nitrated peptides as biomarkers for clinical diagnosis or prognosis, we selected the nitrated peptides that were significantly upregulated in MFS patients (Fig. 6c). The quantitative values of these peptides were combined to obtain a nitrated plasma index (NPI), which provides a measure of the increase in nitration. Mean NPI was clearly increased in the MFS population, with most MFS patients remaining above the mean of the healthy controls (Fig. 6d).

We additionally searched for aortic tissue proteins whose nitration levels could be affected by NO–sGC–PRKG pathway activation. Quantitative proteomics analysis in pooled aorta samples from 12 WT ($n = 6$ pools) and 12 MFS ($n = 6$ pools) mice identified 124 nitrated peptide species from 104 nitro-proteins, which included 147 NS (Supplementary Data 5). Of these NS, 75% correspond to nitro-Tyr and 25% to nitro-Trp. MFS mice showed a consistent and significant upregulation of 24 aortic nitro-proteins (Fig. 6e and Supplementary Fig. 8). This analysis revealed a sharp increase of Acta2 nitration as well as substantial increases in the nitration of seven additional

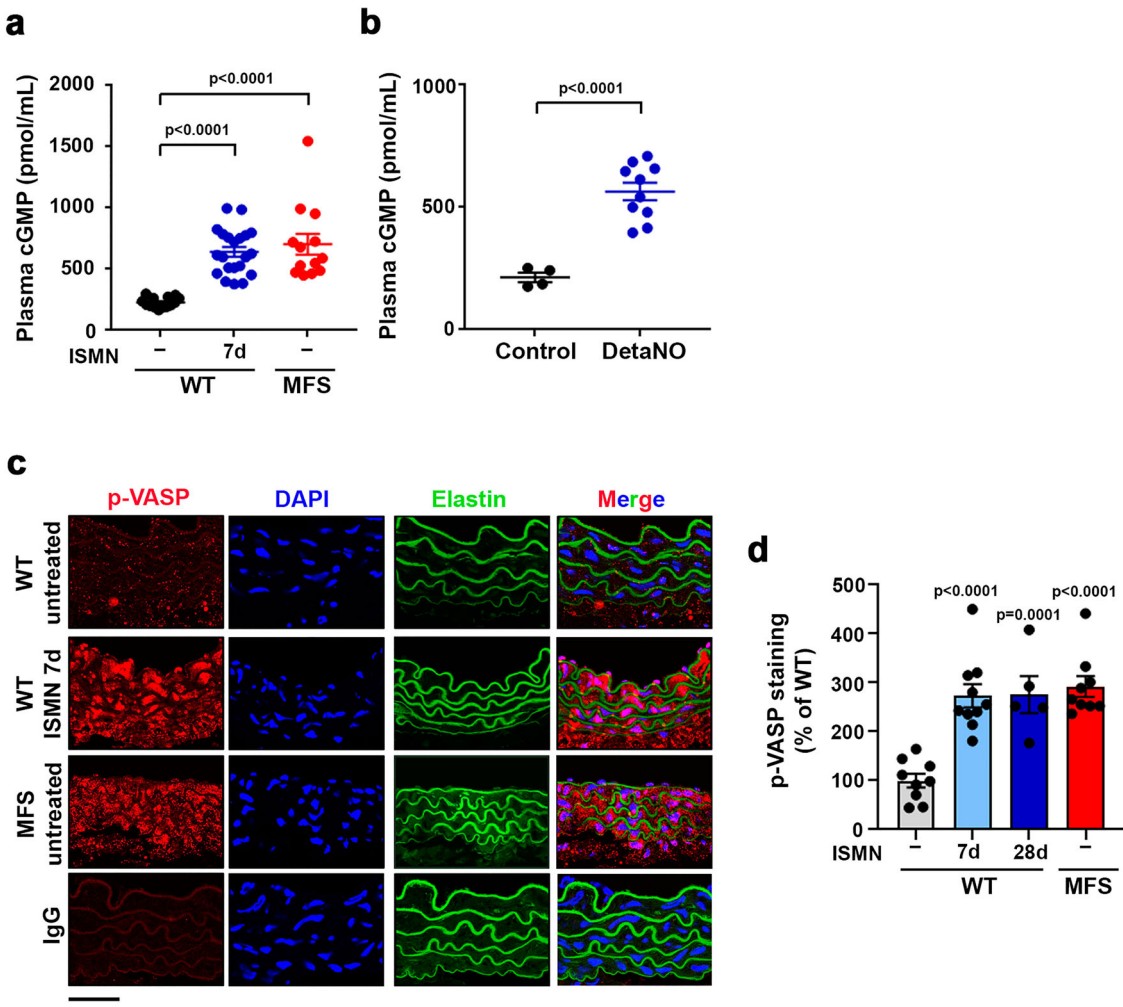

**Fig. 4 The sGC–PRKG pathway is activated in MFS and in NO-donor-treated mice. a** Plasma cGMP in 12–13-week-old untreated WT mice (−) ($n = 14$), MFS mice ($n = 13$), and WT mice treated with 50 mg/kg/day ISMN for 7 d ($n = 20$). **b** Plasma cGMP in untreated WT mice (Control) ($n = 4$) and WT mice treated with 5 mg/kg/day DetaNO for 2 d ($n = 10$). **c** Representative images of pVASP-S239 immunofluorescence (red), elastin autofluorescence (green), and DAPI-stained nuclei (blue). **d** Quantification of pVASP-S239 immunofluorescence in aortic sections from 12-week-old untreated WT mice (−) ($n = 9$), MFS mice ($n = 9$), and WT mice treated with 50 mg/kg/day ISMN for 7 d ($n = 10$) or 28 d ($n = 5$). IgG staining served as a negative control. Scale bar, 50 μm. **a, b, d** Data are shown relative to untreated WT mice as mean ± s.e.m. Data points denote individual mice. Differences were analyzed by **a, d** one-way ANOVA with Dunnett's post-hoc test or **b** unpaired two-tailed $t$-test ($p$-values are shown). Source data are provided in the Source Data file.

cytoskeletal and extracellular matrix proteins that might play a role in contractility regulation (Fig. 6f).

**The NO–sGC–PRKG pathway mediates aortopathy in MFS**. Activation of the NO–sGC–PRKG signaling pathway in MFS patients and mice suggested its involvement in MFS aortopathy. However, increased plasma-protein nitration in MFS patients and mice is also consistent with a cGMP-independent contribution of NO to aortic disease. To ascertain whether sGC–PRKG activation plays a causal role in MFS aortopathy, we treated MFS mice with pharmacological inhibitors of these enzymes. Daily intraperitoneal (i.p.) administration of the sGC inhibitor ODQ (20 mg/kg/day) to 14-week-old MFS mice (Fig. 7a) markedly decreased plasma cGMP (Fig. 7b), inhibited Prkg activity determined by pVASP-S239 aortic staining (Fig. 7c, d), and completely reversed AsAo and AbAo dilation after 7 days without inducing arterial hypertension (Fig. 7e, f). Histological analysis of AsAo cross-sections after 21 days of treatment showed that pharmacological sGC inhibition partially reverted elastic-fiber fragmentation and decreased aortic wall thickness (Supplementary Fig. 11).

Daily i.p. treatment of MFS mice with the PRKG inhibitor KT5823 (2 μmol/kg/day) (Fig. 7g) had similar effects on Prkg activity (Fig. 7h, i), AsAo and AbAo dilation, and BP (Fig. 7j, k), suggesting that sGC and PRKG mediate aortic disease in MFS. However, no reversion of medial degeneration was observed after 7 days of KT5823 treatment (Supplementary Fig. 11a), suggesting that tissue regeneration might require a longer period of Prkg suppression.

To test this hypothesis, we silenced *Prkg* expression in the aortas of adult MFS mice. There are 2 Prkg isoforms: Prkg1, which is ubiquitously expressed but particularly abundant in vascular smooth muscle cells; and Prkg2, which is mainly expressed in non-vascular tissues[47]. We therefore screened candidate shRNAs specific for *Prkg1* in cultured VSMCs, identifying shPrkg1-A and shPrkg1-B as having high silencing capacity (Fig. 8a). The shRNA-encoding lentivirus also encoded green fluorescent protein (GFP) to facilitate assessment of transduction efficiency[48,49]. In vitro transduction of MFS VSMCs with control shRNA (shScr), shPrkg1-A, or shPrkg1-B lentivirus yielded similar GFP mRNA expression levels, and *Prkg1* expression was specifically silenced by shPrkg1-A and shPrkg1-B lentiviruses (Supplementary Fig. 12a).

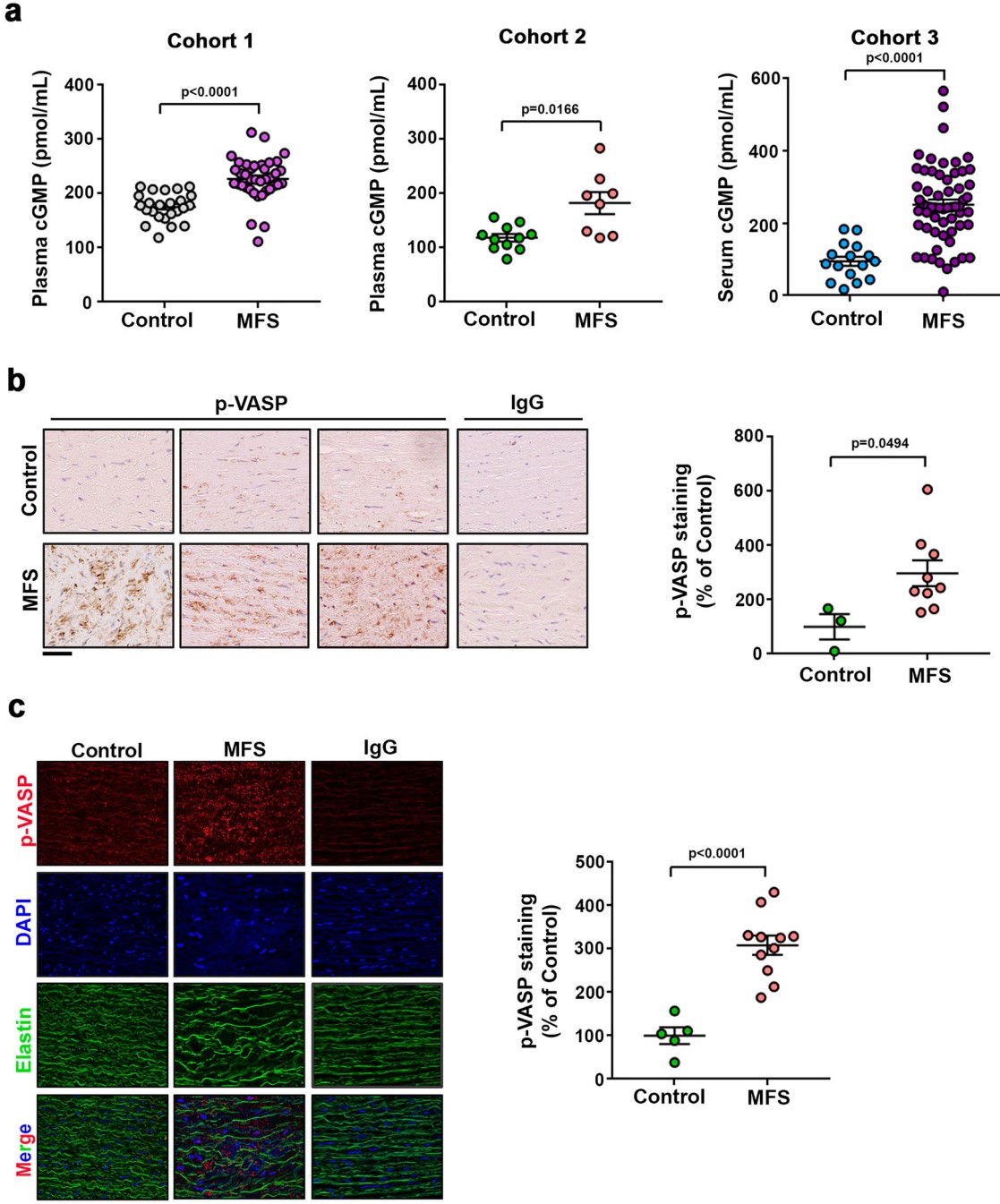

**Fig. 5 The sGC–PRKG pathway is activated in aortas of MFS patients. a** Plasma (Cohorts 1 and 2) and serum (Cohort 3) cGMP in three independent cohorts including 24 healthy donors and 38 MFS patients (Cohort 1); 11 healthy donors and 8 MFS patients (Cohort 2); and 11 healthy donors and 33 MFS patients (Cohort 3). Data are mean ± s.e.m. Each data point denotes an individual. **b** Representative medial layer images and quantification of pVASP-S239 immunohistochemistry in aortic cross-sections of human samples from 3 control donors and 9 MFS patients. Scale bar, 50 μm. IgG staining served as a negative control. Data are shown relative to healthy donors as mean ± s.e.m. Each data point denotes an individual. **c** Representative medial layer images and quantification of pVASP-S239 immunofluorescence (red) and DAPI-stained nuclei (blue) in sections from 5 control donors and 11 MFS patients. Scale bars, 50 μm. IgG staining served as a negative control. Data are shown relative to healthy donors as mean ± s.e.m. Each data point denotes an individual. **a–c** Differences were analyzed by unpaired two-tailed *t*-test with Welch's correction (*p*-values are shown). Source data are provided in the Source Data file.

Consistent with the results obtained upon pharmacological Prkg inhibition, Prkg1 knockdown decreased the expression of the contractility markers *Acta2*, *Tagln2*, and *Cnn1* in MFS cells to normal levels (Supplementary Fig. 12a) while increasing actin fiber formation to normal levels (Supplementary Fig. 12b). Prkg1 knockdown did not, however, restore the capacity of MFS VSMCs

to generate extracellular fibrillin-1 fibers (Supplementary Fig. 12c). Intrajugular delivery of lentivirus encoding shPrkg1-A or shPrkg1-B into MFS mice (Fig. 8b) yielded efficient transduction of all aortic layers, determined by GFP immunostaining of aortic sections 4 weeks after lentivirus delivery (Fig. 8c). Prkg1 protein expression was almost undetectable in aortic samples from these mice (Fig. 8c,

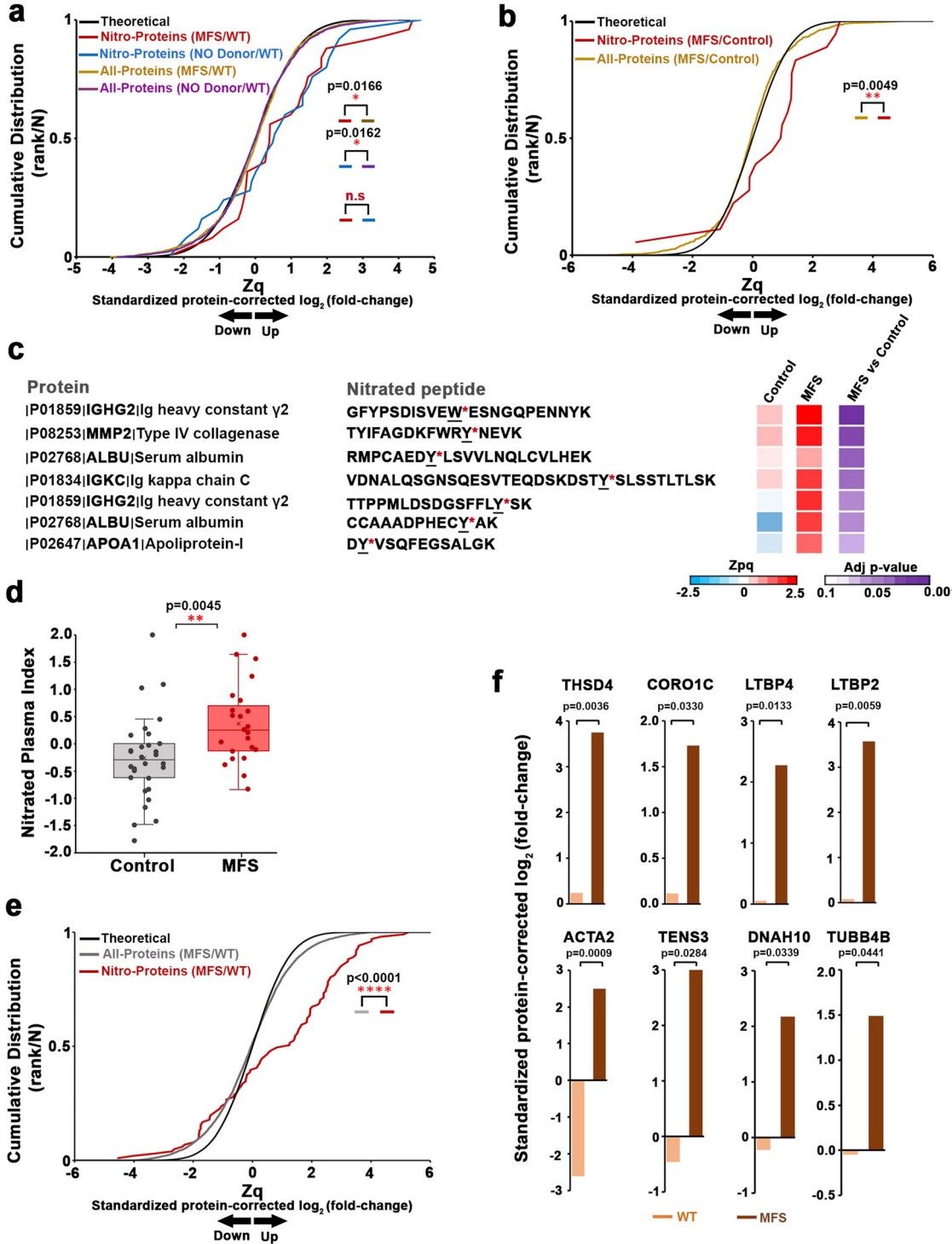

d). Consistent with the results obtained with the PRKG inhibitor, *Prkg1* silencing in the aortas of MFS mice markedly decreased pVASP-S239 aortic staining (Fig. 8e, f) and reversed AsAo and AbAo dilation (Fig. 8g). *Prkg1* knockdown did not raise BP above normotensive values in untreated littermate controls (Fig. 8h). Histological analysis of aortic cross-sections showed that Prkg1 silencing for 28 days led to an almost complete reversal of medial degeneration, determined by the thickness of the aortic wall (Fig. 8i, j) and the regression of elastic-fiber fragmentation and disarray (Fig. 8i, k). Together, these results urge the evaluation of sGC and PRKG as potential targets for therapeutic intervention.

## Discussion

Our results demonstrate that the NO–sGC–PRKG signaling pathway mediates aortopathy in a mouse model of MFS and is activated in MFS mice and MFS patients. These findings identify potential targets for intervention in human MFS, as well as circulating activation markers of this pathway that might be useful for MFS disease monitoring and clinical follow-up.

Our previous work in MFS mice showed that NOS2, whose expression is induced in VSMCs from MFS mice and patients, is an important mediator of medial degeneration and aortic dilation, key features of MFS aortopathy that are efficiently reverted by NOS2 inhibition[19]. NOS2 activity generates much larger

**Fig. 6 Quantitative proteomics shows that increased protein nitration is a signature of MFS in mice and humans.** Quantitative proteomics analysis of plasma samples from **a** untreated WT mice ($n = 7$), MFS mice ($n = 7$), and DetaNO-treated WT mice ($n = 6$) and **b** MFS patients ($n = 23$) and healthy donors ($n = 30$). Data were analyzed using the SanXoT package. Theoretical, normal distribution; Nitro-proteins, cumulative distributions of indicated nitrated protein values corrected to the values for nonmodified peptides from the same protein; All-proteins, cumulative distributions of nonmodified peptides. Data are expressed in standardized log2-ratios (Zq) relative to **a** untreated WT or **b** healthy donors (Control). Differences were analyzed by two-tailed Kolmogorov-Smirnov test (p-values are shown; n.s., not significant). Representative annotated fragmentation spectra for 2 mouse nitro-peptides are provided in Supplementary Fig. 9. All MS/MS spectra of identified tryptic nitro-peptides are provided in the Data availability section. **c** List of nitrated peptides showing significantly higher abundance in MFS patients. The heatmap shows standardized log2-ratios (Zpq) and statistical significance calculated using limma analysis. The annotated fragmentation spectra of these 7 nitro-peptides are provided in Supplementary Fig. 10. **d** Nitrated plasma index, defined as the weighted mean of the nitro-peptides listed in **c**, for 23 MFS patients and 30 healthy donors (Control). Each data point denotes an individual, boxes enclose the interquartile range (IQR), the line in the box shows the centre (median), and whiskers extend 1.5 times above and below the IQR. Differences were analyzed by unpaired two-tailed Student's t-test. **e** Quantitative proteomics analysis of pooled aortic samples from 12 untreated WT mice ($n = 6$ pools) and 12 MFS mice ($n = 6$ pools). Nitrated protein distributions are shown as in **a** and **b**. Differences were analyzed by two-tailed Kolmogorov-Smirnov test. **f** Barplots showing standardized log2-ratios (Zq) integrated by condition (untreated WT and MFS) for significantly upregulated nitro-proteins related to the cytoskeleton: thrombospondin type-1 domain-containing protein 4 (THSD4), coronin 1-C (CORO1C), latent-transforming growth factor beta 2 and 4 (LTBP2, LTBP4), actin aortic smooth muscle (ACTA2), tensin 3 (TNS3), dynein heavy chain 10 (DNAH10), and tubulin beta 4 chain (TUBB4B). Differences were analyzed by limma statistical analysis (exact p-values are shown). Source data are provided in the Source Data file.

amounts of NO than NOS3 or NOS1[38], and therefore its induction in VSMCs might lead to the sustained production of high NO levels. This may in turn activate a number of signaling pathways, with the major candidate being the sGC–PRKG pathway. However, NO can also signal through sGC–PRKG-independent pathways, including those involving nitrosative stress[44].

Our results indicate that supraphysiological NO levels, generated by administration of ISMN or DetaNO, suffice to trigger an MFS-like aortopathy in WT mice. These NO donors induce dose-dependent aortic dilation, elastic-fiber fragmentation, and a drop in BP, as well as recapitulating the sGC–PRKG signaling activation seen in MFS. The association of increased NO–sGC–PRKG signaling with aortopathy warrants consideration of the potential detrimental effects of drugs in clinical use that elicit chronic stimulation of this pathway. This applies not only to NO donors, but also to drugs that impair cGMP degradation, including phosphodiesterase type 5 (PDE5) inhibitors. NO donors have been extensively used to treat angina pectoris and heart failure, and PDE5 inhibitors to treat pulmonary hypertension and erectile dysfunction, with, to our knowledge, no reported pathological consequences related to aortic homeostasis or dilation. Based on accumulated clinical experience, we do not anticipate that patients treated with these agents will develop harmful aortic dilation. Nevertheless, our results with high NO-donor doses in mice suggest that these patients may be at risk of mild aortic dilation. Our results therefore indicate the need of caution with drugs that chronically activate the NO–sGC–PRKG signaling pathway, since they may be detrimental to aortic homeostasis. These drugs include advanced sGC stimulators such as Vericiguat (BAY 1021189) and Riociguat (BAY 63–2521), the latter having received FDA approval for pulmonary hypertension and heart failure[50,51], as well as novel short-acting sGC activators that lack hypotensive side effects, such as TY-55002, a candidate for the treatment of patients with acute decompensated heart failure[52].

Seemingly in conflict with our results, increased NO production by constitutively active Nos3 has been reported to exert a beneficial effect on the aorta in MFS mice[53]. It should be noted, however, that NO levels generated by Nos3 are almost 1000-fold lower than those generated by Nos2[38]. Given that NOS2 inhibitors regress aortic disease in MFS mice[19], our data suggest that the overproduction of NO by NOS2 underlies MFS aortopathy through sGC–PRKG pathway activation. It therefore seems that Nos2 activity might overcome the potential positive effects of Nos3 in MFS aorta.

When uncoupled, NOS2 can lead to overproduction of superoxide anion and peroxynitrite, which activate many signaling pathways[44,54]. NOS2-mediated ROS overproduction can trigger a positive feedback loop via oxidation of the NOS cofactor BH4[55], further uncoupling NOS2 and increasing superoxide production. In this scenario, high NO and superoxide anion levels would generate peroxynitrite, leading to nitrosative stress accompanied by Tyr/Trp-nitration and S-nytrosylation of proteins and oxidative damage to other biomolecules[18]. Oxidative stress has been linked to syndromic and nonsyndromic familial aortic diseases[28,56]; moreover, in MFS mice redox stress is associated with NOX4 (NADPH oxidase 4), which is upregulated in the aortas of MFS patients and whose deficiency protects MFS mice from elastic-fiber fragmentation and aortic dilation[28].

Because NO diffuses from the vascular wall to the lumen, the rise in nitrated proteins in MFS mice and patients indicates increased NO production in MFS and suggests that, despite the presence of high ROS levels, NOS2 generates high amounts of NO in MFS mice and patients. These data are also consistent with a recent report showing significant elevation of NO-derived metabolites such as nitrites in the plasma of MFS patients[57].

Nitration on Tyr and Trp is considered a stable NO-derived post-translational modification[58], and only a few studies have reported these modifications in plasma[59–61]. Given the low abundance of nitration, its detection in complex samples such as plasma is challenging with currently available technology, and we were able to detect Tyr/Trp-nitration only in some abundant proteins. Nevertheless, our data show that nitration is markedly upregulated in MFS mice and that MFS mice and NO-treated WT mice have similar nitration profiles, further supporting the notion that excessive NO production underlies MFS. Moreover, given the upregulation of plasma-protein nitration in MFS patients, our data also suggest that a highly nitrated plasma profile may be a signature of MFS, warranting future assessment of this post-translational modification as a biomarker of the disease.

Other markers of MFS identified in our study include cGMP, whose plasma levels are substantially upregulated in MFS mice and three MFS patient cohorts. Circulating cGMP appears to be a good indicator of sGC activity that can be easily determined in patient plasma or serum. Indeed, elevated cGMP has been reported in other diseases, including congestive heart failure and several cancers[62–64].

Given that pVASP-S239 appears to be a good marker of PRKG activity[30], its increase in the aortas of MFS patients strongly suggests enhanced PRKG activity in this tissue. This idea is supported by the marked drop in pVASP-S239 in MFS VSMCs treated with the PRKG inhibitor KT5823 and in the aortas of MFS mice treated with ODQ, KT5823, or lentivirus encoding

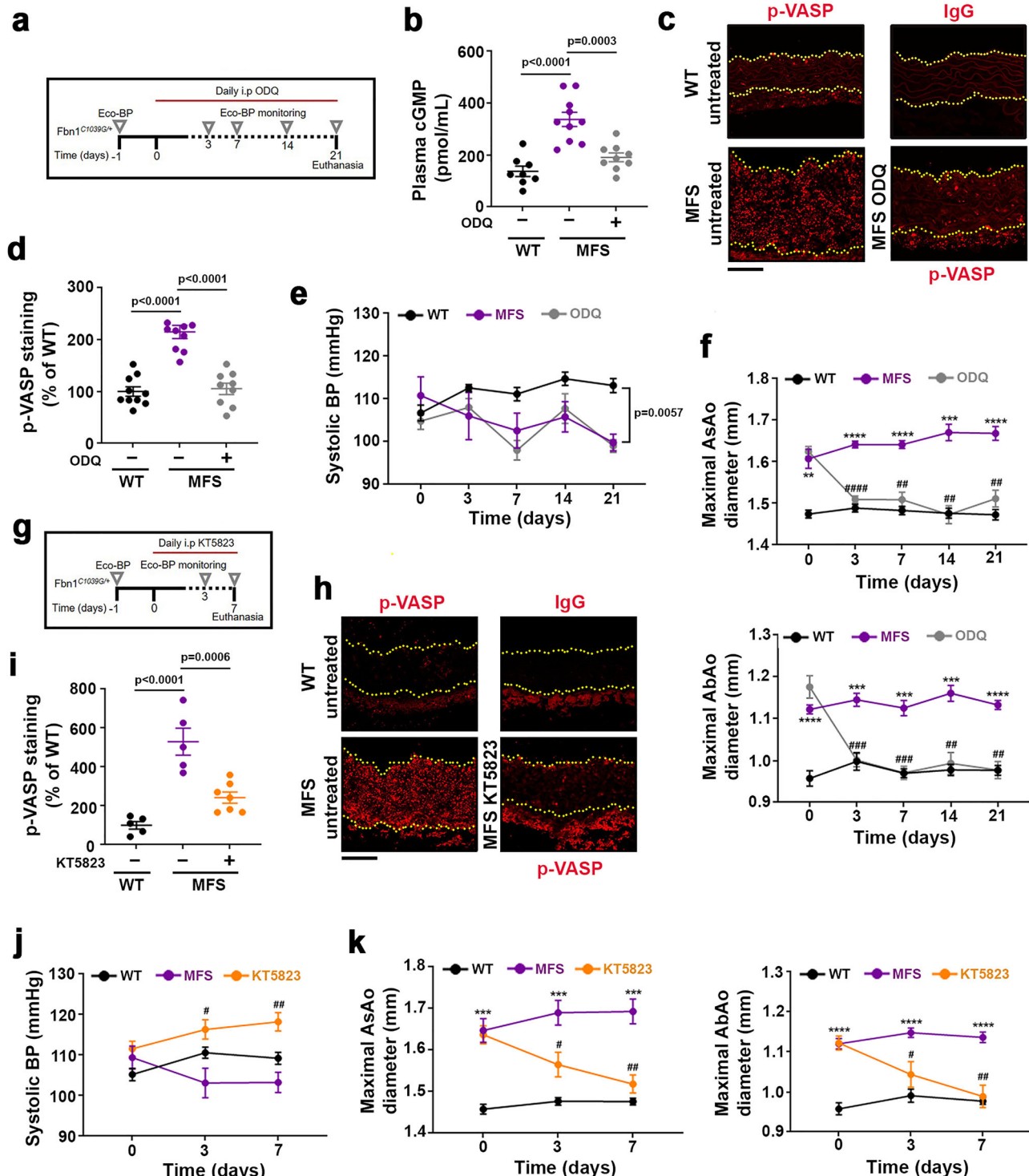

*Prkg1*-shRNA. These findings suggest that analysis of pVASP-S239 in TAAD tissue samples obtained during elective or emergency aortic root surgery could be used to assess the extent of PRKG activation in these tissues.

Aortic dissection is the major cause of morbidity and mortality in MFS. Unfortunately, dissection often occurs when the aortic diameter is below the recommended threshold for elective prophylactic surgery[65,66]. This is also the case for type B dissections in MFS patients with prior prophylactic aortic surgery[67]. The identification of biomarkers that predict the risk of dissection would help in surgical decision-making. Plasma cGMP and Tyr/

Trp-nitrated proteins could help to monitor the course of the disease or the efficacy of future treatments in clinical trials; however, future studies involving a larger clinical sample will be needed to support the prognostic potential of plasma cGMP and protein nitration. It is important to interpret changes in the identified markers with caution because our comparison of circulating markers was limited to MFS patients and healthy donors. Upregulated levels of the markers we describe here are likely a feature of other diseases involving NO pathway activation, such as infection, inflammation, allergy, or sepsis[68,69]. Although the identity of nitrated plasma proteins may differ depending on the

**Fig. 7 Pharmacological inhibition of sGC or PRKG reverts aortopathy in Marfan syndrome. a** Experimental design. 14-week-old MFS mice were treated daily for 21 days with 20mg/kg/day ODQ. Longitudinal ultrasound and BP analysis (Eco-BP) was performed at the indicated times (empty triangles). **b** Plasma cGMP at 21 d ($n = 8$ WT mice, $n = 10$ MFS mice, and $n = 9$ ODQ-treated MFS mice). **c** Representative pVASP-S239 immunofluorescence (red) in mouse aortic sections. Yellow dashed lines delineate the lumen boundary. IgG staining served as a negative control. Scale bar, 50 μm. **d** Quantification of pVASP-S239 immunofluorescence in aortic sections from 10 untreated WT mice (−), 10 MFS mice, and 9 MFS mice treated with 20 mg/kg/day ODQ for 21 d. IgG staining served as a negative control. Scale bar, 50 μm. **b, d** Data are mean ± s.e.m. Each data point denotes an individual mouse. Differences were analyzed by one-way ANOVA with Tukey's post-hoc test (p-values are shown). **e, f** Systolic BP (**e**) and maximal AsAo and AbAo (**f**) diameter at the indicated times ($n = 5$ mice each for untreated and ODQ-treated MFS groups; $n = 7$ mice for WT group). Data are mean ± s.e.m. **$P < 0.01$, ***$P < 0.001$, and ****$P < 0.0001$ (versus WT mice); ##$P < 0.01$, ###$P < 0.001$, and ####$P < 0.0001$ (versus untreated MFS) by repeated-measurements two-way ANOVA with Tukey's post-hoc test. **g** Experimental design. 14-week-old MFS mice were treated daily for 7 days with 2 μmol/kg/day KT5823 and monitored for aortic dilation and BP before treatment and 3 d and 7 d post-treatment. **h** Representative pVASP-S239 immunofluorescence (red) in mouse aortic sections. Yellow dashed lines delineate the lumen boundary. IgG staining served as a negative control. Scale bar, 50 μm. **i** Quantification of pVASP-S239 immunofluorescence in aortic sections from untreated WT mice (−) ($n = 5$), MFS mice ($n = 5$), and MFS mice treated with 2 μmol /kg/day KT5823 for 7 d ($n = 7$). Data are shown relative to untreated WT mice as mean ± s.e.m. Each data point denotes an individual mouse. Differences were analyzed by one-way ANOVA with Tukey's post-hoc test (p-values are shown). **j, k** Systolic BP (**j**) and maximal AsAo and AbAo (**k**) diameter at the indicated times ($n = 10$ for untreated WT group; $n = 8$ per each MFS group). Data are mean ± s.e.m. ***$P < 0.001$ and ****$P < 0.0001$ (versus WT), #$P < 0.05$ and ##$P < 0.01$ (versus untreated MFS) by repeated-measurements two-way ANOVA with Tukey's post-hoc test. Source are provided in the Source Data file.

pathology, these markers would in principle be more suitable for monitoring or predicting disease course than for diagnosis. Again, future studies in larger clinical cohorts will be important in determining whether pharmacological treatment can modify the profile of nitrated proteins and the levels of cGMP. The presence of increased plasma cGMP and protein nitration in patients and MFS mice indicates that medical treatment does not induce this increase, but it will be important to confirm that medication does not affect the protein nitration profile. Regardless of their potential as biomarkers in familial aortopathies, cGMP and protein nitration appear to be useful tools for determining if this pathway is also activated in other TAADs.

Tissue levels of pVASP-S239 could also be used to determine if PRKG activation is a signature of other TAADs. Recent findings show that a gain-of-function mutation in the human *PRKG1* gene generates a constitutively activated kinase and predisposes to TAA in affected families; moreover, the equivalent activating mutation in heterozygous mice (*Prkg1*[R177Q/+] mice) triggers an age-dependent aortic dilation[56,70,71]. In affected families, the increased PRKG1 activity stimulates myosin regulatory light chain phosphatase, which can alter VSMC contractility[71]. It is thus clear that PRKG activation can mediate aortopathy in syndromic and nonsyndromic familial TAAD. It remains to be seen whether sGC–PRKG pathway activation also mediates disease in syndromic diseases other than MFS or in nonsyndromic TAADs not caused by *PRKG1* mutation.

An essential role of NO–sGC–PRKG signaling in aortopathy does not exclude roles in the disease for NOS2-derived signals independent of this pathway. Indeed, oxidants such as NOS2-derived superoxide anion can also activate PRKG[29], and signals downstream of PRKG involving oxidative stress and JNK activation have been shown to mediate aortopathy in *Prkg1*[R177Q/+] mice, in which antioxidants prevent age-dependent aortic dilation[56]. However, the action of PRKG appears to be mediated mainly through its regulation of contractility via myosin light chain phosphatase activation and decreased calcium influx in VSMCs[44,72]. The drop in cellular calcium levels might contribute to aortic disease, as suggested by the deleterious effect of calcium channel blockers (CCBs) in MFS mice and by the increased risk of aortic dissection and need for aortic surgery in MFS patients treated with CCBs compared with patients receiving other antihypertensive drugs[73]. Indeed, the aortic damage caused by CCBs in MFS might be attributable to its cooperation with PRKG activation in depleting cellular calcium stocks. Guideline recommendations for CCBs as an alternative blood pressure regulator in

patients at risk of aortic aneurysm should perhaps be revisited. Our data showing VASP-mediated modulation of actin fiber formation suggest that PRKG regulation of contractility might be regulated not only through the myosin components of the acto-myosin cytoskeleton, but also through actin. Since actin Tyr nitration impairs actin polymerization dynamics[74], our finding that numerous Acta2 Tyr residues are nitrated in the aortas of MFS mice reveals an additional mechanism for NO-mediated regulation of actomyosin cytoskeleton dynamics tuning and for contractility dysregulation. In this scenario, the induction of contractility markers might be seen as an adaptative response to a deficient capacity of the aorta for contraction.

Our results suggest that the control of NO levels and signaling in the aorta is critical for VSMC homeostasis and implicate abnormally high NO–sGC–PRKG signaling as the underlying cause of aortopathy in MFS mice and patients. PRKG can be activated by superoxide anion; however, our data showing that sGC pharmacological inhibition reduces pVASP-S239 in vitro and in vivo strongly suggest that PRKG activation in MFS is dependent on sGC activity. Furthermore, the regression of aortic dilation in MFS mice treated with ODQ supports a causal role for sGC activation in the disease. Similarly, the reversion of aortic dilation and medial degeneration in MFS mice treated with PRKG inhibitors or aortic *Prkg1* knockdown also strongly suggests that MFS is mediated by PRKG activation.

Although prophylactic surgery has increased the lifespan of MFS patients, there are currently no pharmacological treatments able to arrest aortic growth or prevent dissections[14,75]. There is therefore an urgent need to develop effective therapies. Our data lay the basis for exploring the implication of NO–sGC–PRKG signaling in other forms of TAAD and determining the potential of sGC or PKRG inhibition in the treatment of patients with MFS or other forms of TAAD.

## Methods

**Animal procedures.** Animal procedures and experiments complied with all relevant ethical regulations, were approved by the CNIC Ethics Committee and the Madrid regional authorities (ref. PROEX 80/16) and conformed to EU Directive 2010/63EU and Recommendation 2007/526/EC regarding the protection of animals used for experimental and other scientific purposes, enforced in Spanish law under Real Decreto 1201/2005. Overall mouse health was assessed by daily inspection for signs of discomfort, weight loss, or changes in behavior, mobility, and feeding or drinking habits. Mice were housed in a pathogen-free animal facility under a 12 h light/dark cycle at constant temperature and humidity, and fed standard rodent chow and water ad libitum. *Fbn1*[C1039G/+] mice[33], which harbor a mutation in the *Fbn1* gene, were obtained from Jackson Laboratories (JAX mice stock #012885). This strain had been previously backcrossed to the C57BL/6

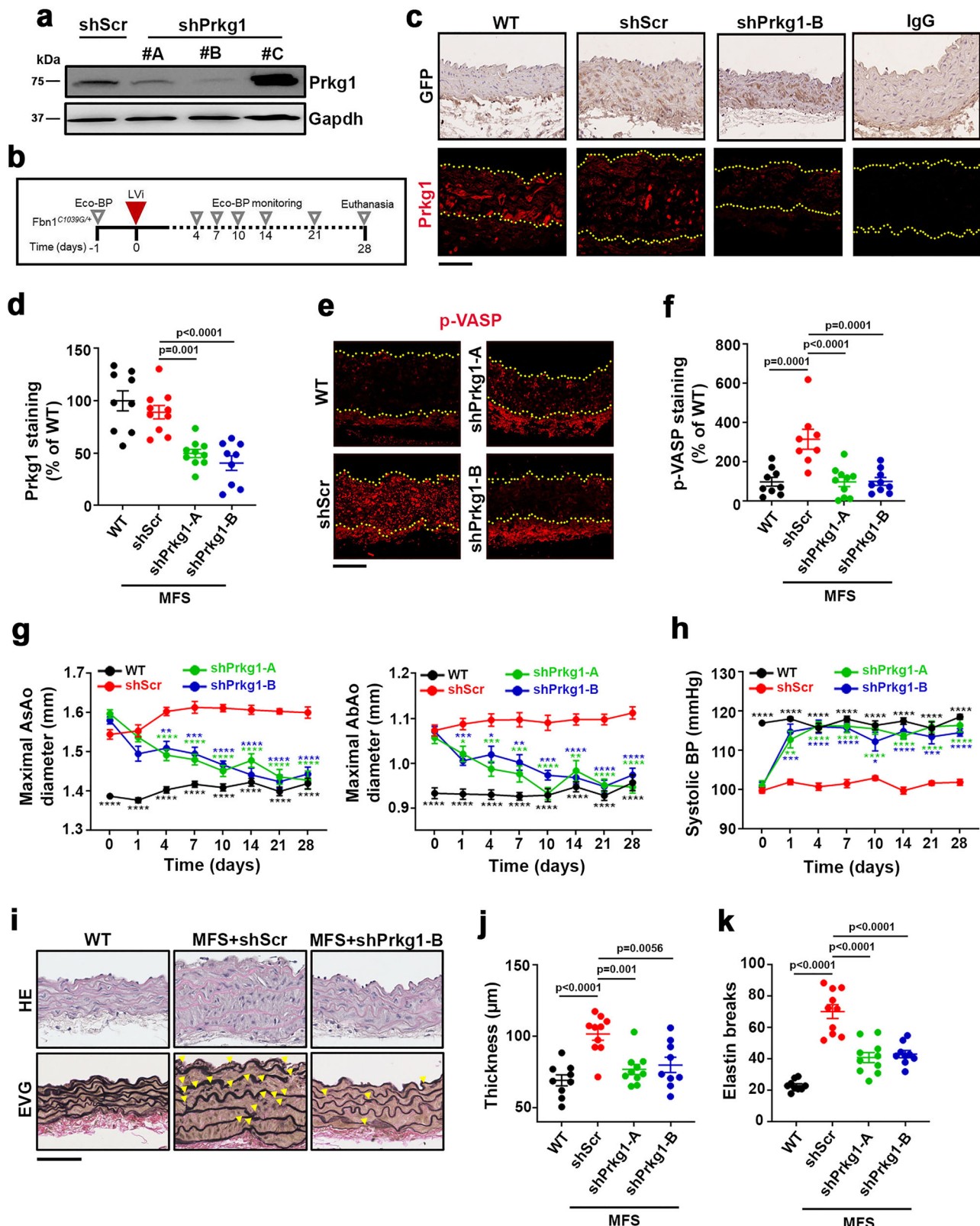

background for more than nine generations. All mice were genotyped by tail-sample PCR using the following primers: 5′-CTCATCATTTTTGGCCAGTTG-3′ and 5′−GCACTTGATGCACATTCACA-3′. Wild-type (WT) mice were on the C57BL/6 background, and 12–15-week-old males and females were used for all experiments. Mice were treated with isosorbide mononitrate (ISMN; MedChem Express; Monmouth Junction, NJ, USA) or DetaNONOate (DetaNO; Enzo Life Sciences; Farmingdale, NY, USA) using subcutaneous osmotic minipumps (Alzet Corp; Cupertino, CA, USA). The specific inhibitors of sGC (ODQ) and PRKG

(KT5823), both from Focus Biomolecules (Plymouth Meeting, PA, USA), were dissolved in dimethylsulfoxide (DMSO), diluted in 0.9% NaCl (DMSO content was <0.5%), and administered i.p. at 20 mg/kg/day for 21 days and 5 mg/kg/day for 7 days, respectively. Control mice received the corresponding intraperitoneal vehicle.

**Blood pressure measurements.** Arterial blood pressure (BP) was measured in mouse tails using the automated BP-2000 Blood Pressure Analysis System and

**Fig. 8 *Prkg1* silencing reverts aortopathy in Marfan syndrome. a** Representative immunoblot analysis ($n = 2$ independent experiments) of Prkg1 expression in WT DetaNO-treated VSMCs transduced with lentivirus (LVi) encoding control shRNA (shScr) or *Prkg1*-specific shRNAs A, B, and C. Uncropped blots in Supplementary Fig. 13. **b** Experimental design; 14-week-old MFS mice were inoculated through the jugular vein with LVi encoding GFP and either a control shRNA (shScr) or *Prkg1*-specific shRNAs A and B. Mice were monitored for aortic dilation and BP prior to treatment and at the indicated times. **c** Representative images of GFP immunohistochemistry (top) and Prkg1 immunofluorescence (bottom). Top and bottom yellow dashed lines delineate the lumen and the adventitia boundary, respectively. IgG staining of AsAo sections from shScr-transduced MFS mice served as a negative control. Scale bar, 50 µm. **d** Quantification of Prkg1 immunofluorescence in AsAo sections of control WT and shScr-, and shPrkg1-B-transduced MFS mice ($n = 9$ in WT mice and MFS mice infected with *shPrkg1*-B, $n = 10$ in MFS mice infected with shScr or *Prkg1*-A). Data are shown relative to WT mice as mean ± s.e.m. Each data point denotes an individual mouse. **e** Representative pVASP-S239 immunofluorescence (red) in mouse aortic sections. Yellow dashed lines delineate the lumen boundary. IgG staining served as a negative control. Scale bar, 50 µm. **f** Quantification of pVASP-S239 immunofluorescence in aortic sections from 9 WT, 8 MFS shScr, 10 MFS shPrkg1-A, and 9 MFS shPrkg1-B mice. Data are shown relative to uninfected WT mice as mean ± s.e.m. Each data point denotes an individual mouse. **d, f** Differences were analyzed by one-way ANOVA with Tukey's post-hoc test (*p*-values are shown). **g, h** Maximal AsAo and AbAo (**g**) diameter and systolic BP (**h**) at the indicated times ($n = 10$ per group). Data are mean ± s.e.m. \*$P < 0.05$, \*\*$P < 0.01$, \*\*\*$P < 0.001$, and \*\*\*\*$P < 0.0001$ versus shScr by repeated-measurements two-way ANOVA with Tukey's post-hoc test. **i** Representative images showing staining with hematoxylin and eosin (HE) and elastic van Gieson (EVG) in AsAo of the indicated mice ($n = 9$–10 mice per group). Yellow arrowheads indicate elastin breaks. Scale bar, 50 µm. **j** Wall thickness and **k** elastin breaks in AsAo sections from the mouse cohorts shown in **i**. Data are mean ± s.e.m. Each data point denotes an individual mouse. **j, k** Differences were analyzed by one-way ANOVA with Tukey's post-hoc test (*p*-values are shown). Source data are provided in the Source Data file.

software (Visitech Systems, Apex, NC, USA)[32,76]. BP measurements were recorded in mice located in a tail-cuff restrainer over a warmed surface (37 °C). Mice underwent a training period, with BP monitored every day for 7 days. After that, baseline BP was measured 1 day before the beginning of the treatment in each mouse cohort. Fifteen consecutive systolic BP measurements were made, and the last 10 readings per mouse were recorded, tested for outliers using the Chauvenet's criterion and averaged. BP monitoring was repeated during experiments as indicated in the Figure legends.

**In vivo ultrasound imaging**. Images of the aorta were obtained in isoflurane-sedated mice (2% isoflurane) by high-frequency ultrasound with a VEVO 2100 echography device (Fujifilm-VisualSonics, Toronto, Canada) with a transducer that provides 30-micron axial resolution. Maximal internal aortic diameters were measured at systole using VEVO 2100 software, version 1.5.0 (VisualSonics). Measurements were taken before treatment initiation to determine the baseline diameters and were repeated several times during the experiment.

**Cell procedures**. Primary mouse vascular smooth muscle cells (VSMCs) were isolated from 4–6-week-old mice and grown as described[31,32]. Briefly, tissue was digested with a solution of collagenase and elastase until a single-cell suspension was obtained. Cells were then cultured at 37 °C, 5% $CO_2$ in growth medium (Dulbecco's modified Eagle's medium [DMEM]) supplemented with 20% fetal bovine serum [FBS]). All experiments were performed during passages 3–7. VSMCs were FBS-starved 36 h before stimulation experiments. The HEK-293T (CRL-1573) and Jurkat (Clone E6-1, TIB-152) cell lines, required for high-titer lentivirus production and lentivirus titration, respectively, were purchased from ATCC. All cells were mycoplasma-negative.

VSMCs were incubated with 100µM DetaNO or 100 µM 8-bromo-guanosine 3′,5′-cyclic monophosphate (8-Br-cGMP; Biolog; Bremen, Germany). In some experiments, the NOS inhibitor N$^G$-nitro-L-arginine methyl ester (L-NAME, 300 µM; Sigma–Aldrich; St. Louis, MO, USA), ODQ (10 µM), or KT5823 (1 µM), were added to VSMCs 1 h before the addition of DetaNO or 8-Br-cGMP. In another set of experiments, MFS VSMCs were incubated for 24 h with KT5823.

For cell immunostaining, cells were fixed with 4% paraformaldehyde for 10 min and permeabilized with 0.3% Triton X-100 in PBS for 30 min. Samples were incubated overnight with anti-phospho-VASP (1:50, sc-101439; Santa Cruz Biotechnology; Santa Cruz, CA, USA) or rabbit anti-fibrillin-1 polyclonal antibody[77] (1:2000, pAb9543; kindly donated by Dr. L. Sakai). The secondary antibody was AlexaFluor568-conjugated goat anti-mouse (1:500; A-11031; Molecular Probes; Carlsbad, CA, USA). To determine F-actin formation, cells were stained with Texas Red-X-conjugated phalloidin (1:2000, T7471; Thermo Fisher Scientific, Bremen, Germany) in at least three independent batches, with three technical triplicates per condition and batch and at least three images captured per triplicate. Images were acquired at 1024 × 1024 pixels, 8 bits, using a Confocal TCS Leica SP5 microscope (Leica Microsystems GmbH; Wetzlar, Germany) fitted with a ×40 oil-immersion objective and Leica LAS-AF V2.7.3. acquisition software. All images were processed for presentation with Photoshop (Adobe) according to the guidelines of this journal and analyzed with ImageJ software (version 1.52a, NIH, http://rsb.info.nih/ij/). Staining was quantified after setting an intensity threshold to include only specific signals. For intracellular staining, total intensity was relativized to cell area. For extracellular staining of Fbn1, the signal was calculated as the stained area. Each condition's value was relativized to the average staining of the experiment.

**Lentivirus production and infection**. A lentiviral plasmid encoding GFP and shRNA targeting mouse *Prkg1* was engineered by cloning the following shRNAs into the pH1-DUAL lentiviral vector: shPrkg1-A (sense) 5′- GATCCCCTTGCTTT GCTCTGATTATACTCGAGTATAATCAGAGCAA AGCAAGGTTTTTTGC-3′; shPrkg1-A (antisense) 5′- GGCCGCAAAAAACCTTGC TTTGCTCTGATTATAC TCGAGTATAATCAGAGCAAAGCAAGGg-3′; shPrkg1-B (sense) 5′- GATCCCC GGAGAATCTCATCCTAGATCTCGAGATCTAGGATGAGA TTCTCCGGTTT TTGC-3′; shPrkg1-B (antisense) 5′- GGCCGCAAAAACCGGAGAA TCTCATCC TAGATCTCGGATCTAGGATGAGATTCTCCGGG. Pseudo-typed lentiviruses were produced by transient calcium phosphate transfection of HEK-293T cells with a plasmid encoding GFP and *Prkg1*-specific shRNA. Viruses were concentrated from culture supernatant by ultracentrifugation (2 h at 121,896xg; Ultraclear Tubes; SW28 rotor and Optima L-100 XP Ultracentrifuge; Beckman, Brea, CA, USA), suspended in cold sterile PBS, and titrated by infection of Jurkat cells. Jurkat cells were seeded in a 96-well plate and infected with the desired lentiviral dilution (1:10-1:100000). After 48 h, cells were centrifuged 5 min at 1800 × *g* and resuspended in 200 microL of ice-cold PBS with 1:1000 propidium iodide. Transduction efficiency (% of GFP-expressing cells) and cell death (propidium iodide incorporation) were quantified by flow cytometry, as illustrated in Supplementary Fig. 14. Flow cytometry data were collected using a Canto 3L HTS cytometer and the BD FACSDiva Software Version 6.1.3, and analyzed using FlowJo 10.7.1 software.

VSMCs were infected (multiplicity of infection = 10) overnight at 37 °C in growth medium. Medium was then replaced with fresh growth medium, and cells were cultured for 7 additional days and then processed for mRNA expression analysis and cell immunostaining or stimulated for 5 min with DetaNO for immunoblot assays. For in vivo transduction experiments, animals were anesthetized (with ketamine and xylazine), and a small incision was made to expose the right jugular vein[49]. Virus solution (100 µl, 10$^9$ particles/ml in PBS) was injected directly into the right jugular vein. Transduction efficiency was analyzed in aortic samples by GFP immunohistochemistry and PRKG immunofluorescence.

**cGMP immunoassay**. cGMP was measured in mouse and human plasma by competitive enzyme immunoassay (KGE003, R&D Systems; Minneapolis, MN, USA). To obtain mouse plasma, blood was extracted after sacrifice by $CO_2$ inhalation using the cardiac puncture method, collected in EDTA tubes, and centrifuged for 15 min at 13,000 × *g*. Plasma and serum samples from MFS patients and healthy blood donors were obtained and processed following standard operating procedures.

**Histology**. After sacrifice of mice by $CO_2$ inhalation, aortas were perfused with saline, isolated, fixed in 10% formalin overnight at 4 °C, and paraffin embedded. Paraffin cross-sections (5 µm) from fixed ascending aortas (AsAo) were prepared for immunochemistry or immunofluorescence, or stained with hematoxylin and eosin, Alcian blue, or modified Verhoeff elastic-Van Gieson (EVG) kit (Sigma–Aldrich). Images were acquired under a Leica DM2500 microscope fitted with a ×40 HCX PL Fluotar objective or with a scanner NanoZoomer-2. ORSC110730. Leica Application Suite V3.5.0 acquisition software and NDP.view 2 V2.7.43 software were used, respectively. Elastic lamina breaks, defined as interruptions in elastic fibers, were counted in the entire medial layer of six non-consecutive cross-sections per mouse, using 4–12 mice per experiment. The mean number of breaks was calculated. The exact number of mice per group is indicated in the figure legends.

For immunofluorescence, deparaffinized sections were rehydrated, boiled 3 min to retrieve antigens in 10 mM citrate buffer containing 0.05% Tween-20, pH6, and blocked for 1 h with 10% goat serum plus 2% BSA in PBS. Samples were incubated with the following antibodies for immunohistochemistry or immunofluorescence: mouse monoclonal anti-pVASP (1:50; sc-101439, Santa Cruz Biotechnology, Santa Cruz, CA, USA), rabbit polyclonal anti-p-VASP (1:25; SAB4300129, Sigma–Aldrich, St. Louis, MO, USA), rabbit polyclonal anti-PRKG (1:50, ADI-KAP-PK005-F, Enzo Life Sciences), and rabbit polyclonal anti-GFP (1:100, A11122, Invitrogen; Carlsbad, CA, USA). Specificity was determined by substituting the primary antibody with unrelated IgG (Santa Cruz) at the same dilutions as the antigen-specific antibodies. For immunohistochemistry, color was developed with DAB (Vector Laboratories; Burlingame, CA, USA), and sections were counterstained with hematoxylin and mounted in DPX (Casa Álvarez, Madrid, Spain). For immunofluorescence, secondary antibodies were polyclonal Alexa-Fluor-647-conjugated goat anti-rabbit or polyclonal Alexa-Fluor-647-conjugated chicken anti-mouse (1:500, Molecular Probes). Sections were mounted with DAPI in Citifluor AF4 mounting medium (Aname; Madrid, Spain). For immunofluorescence and immunohistochemistry in mouse and human aortic tissue, antibody staining was quantified in the medial layer of three nonconsecutive whole cross-sections per aorta, using 4–12 aortas per condition. The rare histological preparations excluded were those lacking a whole aortic cross-section. The whole aortic cross-section was screened before taking representative pictures (minimum of three). Immunostaining was technically validated with samples from the same experimental unit in three independent experiments. Images were acquired at 1024 × 1024 pixels, 8 bits, using a Confocal TCS Leica SP5 microscope (Leica Microsystems GmbH; Wetzlar, Germany) fitted with a 40x oil-immersion objective.

All images were processed for presentation with Photoshop (Adobe) according to the guidelines of this journal and analyzed with ImageJ software (version 1.52a, NIH, http://rsb.info.nih.gov/ij/).

**Real-time and quantitative PCR.** Total RNA was isolated from VSMCs or from aortas by homogenization with TRIzol (Invitrogen; Carlsbad, CA, USA). Total RNA (2 μg) was reverse-transcribed at 37 °C for 50 min in a 20-μl reaction mix containing 200 U Moloney murine leukemia virus (MMLV) reverse transcriptase (Promega, Madison, Wi, USA), 100 ng random primers and 40 U RNase Inhibitor (Invitrogen). The sequences of the sense and antisense primers used for amplification are described in Supplementary Table 1. qPCR reactions were performed in triplicate with SYBR master mix (Promega) according to manufacturer guidelines. Probe specificity was assessed in a post-amplification melting-curve analysis. For each reaction, only one melting-temperature (Tm) peak was produced. The amount of target mRNA in samples was estimated by the $2^{-\Delta CT}$ relative quantification method, using *Gapdh* for normalization. Fold ratios were calculated relative to mRNA expression levels from control animals. qPCR data were analyzed using the Bio-Rad CFX Manager 3.1 software and further analysis was performed in Excel software.

**Immunoblot analysis.** Samples from mouse aortas were isolated, frozen in liquid nitrogen and then homogenized (MagNA lyzer, Roche). VSMCs were washed with ice-cold PBS and lysed in ice-cold total lysis buffer (20 mM Tris-HCl, pH 7.5; 1% Triton X-100; 50 mM NaF; 5 mM MgCl2; 500 mM NaCl; 10 mM EDTA) supplemented with protease, phosphatase, and kinase inhibitors (100 μM benzamidin, 1 μg/ml leupeptin, 1 μg/ml pepstatin, 1 μg/ml aprotinin, 1 μM ditiotreitol, 1 mM PMSF, and 3 mM EGTA). Proteins were separated under reducing conditions on SDS–polyacrylamide gels, transferred to nitrocellulose membranes, and detected with rabbit polyclonal anti-PRKG1 (1:1000, ADI-KAP-PK005-F, Enzo Life Sciences), anti-Gapdh (1:10,000; ab8245 Abcam, Cambridge, UK), rabbit polyclonal anti-GUCY1A3 (1:1000, 12605-1-AP; Proteintech, Rosemont, IL, USA), rabbit polyclonal anti-GUCY1B3 GUCY1A3 (1:500, 19011-1-AP; Proteintech), and mouse monoclonal anti-α-Tubulin (1:40,000, T 6074; Sigma–Aldrich, St. Louis, MO, USA). Bound antibodies were detected with enhanced-chemiluminescence (ECL) detection reagent (Millipore, Darmstadt, Germany). All uncropped blots are presented in Supplementary Fig. 13.

**Human samples.** The study complies with all relevant ethical regulations and was approved by the Research Ethics Committee of Cantabria (ref. 27/2013), the Ethics Committee of Ghent University Hospital (B65020111160), and the Ethics Committee of Instituto de Salud Carlos III (CEI PI91_2018-v2-Enmienda_2019). Aorta, plasma and serum samples from patients with Marfan Syndrome were obtained from biobanks of Hospital Puerta de Hierro, Hospital Vall D'Hebron, and Ghent University Hospital. Tissue samples from MFS patients were obtained during elective or emergency surgery for aortic root replacement. Age- and sex-matched ascending aorta samples used as controls were obtained anonymously from multiorgan transplant donors. During preparation of the heart for transplantation, excess AsAo tissue was trimmed and harvested for the study. Tissues were immediately fixed, kept at room temperature for 48 h, and embedded in paraffin. Age- and sex-matched control plasma samples were obtained from healthy volunteers. Informed consent was obtained from all human participants or their families. Patient clinical data were retrieved while maintaining anonymity.

**Proteomics.** Nitration of plasma and ascending aorta proteins was assessed by high-throughput multiplexed isobaric labeling analysis in control and DetaNO-treated C57BL/6 WT mice and MFS mice. In the plasma experiment, seven control, six DetaNO-treated, and seven MFS animals were used, whereas in the aorta experiment each group of WT and MFS animals consisted of six pools of two biological replicates (a total of 24 animals). In both experiments, samples were analyzed in two TMT 10-plex batches, each containing three or four biological replicates from each experimental condition. In the aorta experiment, two channels were reserved for internal reference standards created by pooling the WT peptide samples. Proteins were extracted from AsAo (50 mg per pool) by tissue homogenization with ceramic beads (MagNa Lyser Green Beads, Roche, Germany) in extraction buffer (100 mM Tris-HCl pH 7.4, 1 mM EDTA, 4% SDS). Plasma proteins (200 μg per animal) were obtained from about 4 μl plasma, which was mixed with the same volume of 50 mM Tris, 2% SDS, and 50 mM DTT. In both experiments, proteins were denatured by boiling for 5 min and subjected to filter-aided digestion (Nanosep Centrifugal Devices with Omega Membrane-10K, PALL)[78]. Briefly, urea was added to each sample to bring SDS concentration down to 0.06%. Samples were then transferred to the filter and centrifuged at 14,000 × g for 10 min. Cys residues were blocked with 50 mM iodoacetamide for 1 h at room temperature in the dark. After three washes with 100 μl 50 mM ammonium bicarbonate pH 8.8, proteins were digested with trypsin (1:30 trypsin:protein, Promega) overnight at 37 °C. After protein digestion, peptides were eluted from the filter in two steps, first with 40 μl 0.5M ammonium bicarbonate and then with 50 μl 0.5M NaCl. Peptides were acidified with trifluoroacetic acid to a final concentration of 1%, desalted on C18 Oasis HLB cartridges (Waters, Milford, MA, USA), and vacuum-dried. Peptides were TMT-labeled according to the manufacturer's instructions and desalted on OASIS extraction cartridges. Plasma peptides were separated into four fractions with increasing amounts of acetonitrile (Fr1, 15%; Fr2, 17.5%; Fr3, 20%; and Fr4, 50%), whereas the aorta peptides where separated into five fractions (Fr1, 12.5%; Fr2, 15%; Fr3, 17.5%; Fr4, 20%; and Fr5, 50%) using the high pH reversed-phase peptide fractionation kit (Thermo Fisher Scientific, Bremen, Germany), and dried-down before MS analysis.

Quantitative proteomics using multiplexed isobaric labeling was also used to analyze nitration of plasma proteins from 30 healthy donors and 23 MFS patients. Samples were analyzed in seven TMT batches. Each TMT batch contained plasma samples from each of the two experimental conditions and two channels were reserved for reference internal standards created by pooling the control peptide samples. Protein digestion and TMT labeling and fractionation of peptides were performed as described above, using five fractions with increasing amounts of acetonitrile.

Each fraction of the labeled peptide samples was analyzed using a Proxeon Easy nano-flow HPLC system (Thermo Fisher Scientific) coupled via a nanoelectrospray ion source (Thermo Fisher Scientific) to either a Q-Exactive HF mass spectrometer (Termo Fisher Scientific) or an Orbitrap Fusion mass spectrometer (Thermo Fisher Scientific). A C18-based reverse phase separation was used, with a 2-cm trap column and a 50-cm analytical column (75 μm I.D, 2 μm particle size, Acclaim PepMap RSLC, 100 C18; Thermo Fisher Scientific). Peptides were loaded in buffer A (0.1% formic acid in water (v/v)) and eluted with an acetonitrile gradient consisting of 0–30% buffer B (80% acetonitrile, 0.1% formic acid) for 300 min and 50–90% B for 3 min at a flow rate of 200 nl/min. Mass spectra were acquired in a data-dependent manner, with an automatic switch between MS and MS/MS using a top-15 acquisiton method with dynamic exclusion. MS spectra were acquired with a mass range of 400–1500 *m/z* and 120,000 FT resolution. HCD fragmentation was performed at 30% normalized collision energy, and MS/MS spectra were analyzed at 30,000 resolution in the orbitrap. Proteins were identified with the SEQUEST HT algorithm integrated in Proteome Discoverer 2.1 (Thermo Fisher Scientific) against a mouse or a human database. The mouse reference proteome database (UniProtKB/Swiss-Prot and TrEMBL December_2016) was supplemented with 116 CRAP proteins (Global Proteome Machine) (50915 sequences) and was concatenated with the inverted database constructed from the same target databases (101830 sequences in total) using DecoyPYrat[79]. The human reference proteome database (UniProtKB/Swiss-Prot July_2018) was concatenated with the corresponding inverted database following the same approach (81632 sequences in total). For database searching, parameters were selected as follows: trypsin digestion with two maximum missed cleavage sites, precursor mass tolerance of 2 Da, and a fragment mass tolerance of 30 ppm. Met oxidation (15.994915 Da), Trp-nitration (44.985078 Da), and Tyr nitration (44.985078 Da) were set as variable modifications. Fixed modifications were 57.021464 Da in Cys and 229.162932 Da in Lys and in the peptide N-terminal position. The false discovery rate (FDR) was calculated based on the search of results against the corresponding decoy database using the refined method[80] with an additional filter for precursor mass tolerance of 15 ppm[81] and estimation of the corrected Xcorr[82]. An FDR of 1% was used as the criterion for peptide identification.

Quantitative information from TMT reporter intensities was integrated from the spectrum level to the peptide level and then to the protein level based on the WSPP model[83,84] using the GIA integration algorithm[85]. Quantitative results were expressed as $\log_2(A_i/C)$, where $A_i$ is the intensity of the TMT reporter of the corresponding sample *i* and *C* is the mean intensity of the TMT reporters from the controls (for mouse plasma samples) or the mean value of the two internal reference standards (for human plasma or mouse aorta samples). The algorithm

was modified to incorporate the quantitative values of modified peptides as part of the automated workflow[85–87]. Hence, quantitative peptide values are referred to the weighted averages of the nonmodified peptides from the same protein and are therefore not affected by protein abundance changes. This statistical model accurately describes the error distribution of abundance changes for both nonmodified and modified peptides in null-hypothesis experiments[86,87]. Peptide quantifications from the animals in the same group were further integrated to obtain an averaged value per condition using the GIA algorithm[83]. Nitro-protein abundance is calculated as a weighted average of all quantified nitrated peptides, which as commented above, are referred to the weighted averages of the nonmodified peptides of the same protein. Relative changes in peptide and protein abundance (log2-ratios) were expressed in standardized units ($Z_{pq}$ and $Z_q$)[83]. Cumulative distributions from two conditions were judged statistically different according to the two-tailed Kolmogorov-Smirnov test. For human samples, abundance differences of nitrated peptides between healthy individuals and MFS patients were analyzed using limma[88]. Plasma-protein nitration in each patient was measured as the nitrated plasma index (NPI), defined as the weighted mean of seven most upregulated nitro-peptides. The statistical difference between mean NPI values for healthy donors and MFS patients was calculated by unpaired two-tailed Student's $t$-test. For mouse aorta samples, abundance differences in nitro-proteins between untreated WT and MFS mice were also analyzed by limma workflow. All tryptic nitrated sequences were annotated in silico by extracting information from the following databases: Swiss-prot, TrEMBL, Nucleotide database and PDB. The assignation of leucine and isoleucine residues in nitrated peptides to the corresponding protein sequences was checked manually.

**Statistical analysis**. Statistical comparisons were carried out with GraphPad Prism software 7.05. Appropriate tests were chosen according to the data distribution. Differences were analyzed by the Student or the Welch $t$-test (depending on the variance of the tested conditions after F-test); one-way, two-way or repeated-measurement two-way analysis of variance (ANOVA); and Bonferroni, Sidack, Tukey, Dunnett, or Newman post-hoc tests (experiments with ≥3 groups), as appropriate (variance equality was checked by the Bartlett test). The D´Agostino-Pearson and Shapiro-Wilk normality tests were applied before assuming normality distribution of the data. Statistical methodology corresponding to proteomics analysis is described above. Statistical significance was assigned at *$P <$ 0.05, **$P < 0.01$, ***$P < 0.001$, and ****$P < 0.0001$.

Sample size was chosen empirically based on our previous experience in the calculation of experimental variability. The initial estimation of sample size was made by using software Gpower3.1.94. After this estimation, the initial number of animals was reduced based on the results obtained from previous experiments. No data were excluded. The numbers of animals used are described in the corresponding figure legends. All experiments were performed with at least three biological replicates, and all attempts on replication were successful. No randomization was performed to allocate animals into experimental groups, and investigators were not blinded to group allocation during in vivo experiments. Experimental groups were balanced in terms of animal age, weight and AsAo/AbAo basal diameter. Male mice were used throughout the manuscript, except for experiments corresponding to Fig. 7, where male and female mice were used. Animals were genotyped before experiments, and they were all caged together and treated in the same way.

**Reporting summary**. Further information on research design is available in the Nature Research Reporting Summary linked to this article.

## Data availability

Source data are provided with this paper for all figures. The mass spectrometry raw data that support the findings of this study, including nitrated peptide MS/MS spectra (Supplementary Data 2 and 4), mouse reference proteome database December 2016 and human reference proteome database July 2018, are publicly available in the Peptide Atlas repository (http://www.peptideatlas.org/PASS/PASS01528). The following databases were used: UniProtKB, Swiss-Prot and TrEMBL (https://www.uniprot.org/uniprot/), Nucleotide database (https://www.ncbi.nlm.nih.gov/nucleotide/), Protein Data Bank, PDB (https://www.rcsb.org/). Other relevant datasets generated and/or analyzed during the current study are available from the corresponding authors on reasonable request. Source data are provided with this paper.

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

## Acknowledgements

We thank L. Sakai for reagents; S. Bartlett for English language editing; S. Lamas and M.A. Hurlé for critical reading of the manuscript and continuous advice; J. Oller for suggestions and help at the beginning of this work; C. Salas for help organizing clinical database and patient samples; N. Martín-Cofrares and V. Labrador for advice on confocal imaging and immunohistochemistry experiments; the CNIC Facilities of histology, proteomics, and advanced imaging; and R. Magni, A.I. Torralbo, and A. Colmenar for excellent technical support and advice. The CNIC is supported by the Spanish Ministerio de Ciencia e Innovación and the Pro-CNIC Foundation and is a Severo Ochoa Center of Excellence (SEV-2015-0505). The project leading to these results has received funding from "La Caixa" Banking Foundation under project codes HR18-00068 (to M.R.C. and J.M.R.) and HR17-00247 (to J.V.); Spanish Ministerio de Ciencia e Innovación grants RTI2018-099246-B-I00 (MICIU/AEI/FEDER, UE) and SAF2015-636333R (MINECO/FEDER, UE) to J.M.R., SAF2017-88881R (MINECO/AEI/FEDER, UE) to M.R.C., and BIO2015-67580-P and PGC2018-097019-B-I00 to J.V.; the Comunidad de Madrid

through the European Social Fund (ESF)-financed program AORTASANA-CM (B2017/BMD-3676) to J.M.R., M.R.C., and A.F.; the Instituto de Salud Carlos III (CIBER-CV CB16/11/00264 and CB16/11/00277; PRB3-IPT17/0019-ProteoRed to J.V.; and PI17/00381, with cofinancing from the European Regional Development Fund, to G.T.-T.); Fundacio La Marato TV3 (20151330 to J.M.R. and A.E., and 122/C/2015 to J.V.); The Marfan Foundation USA Faculty grant 2017 MRF/1701 (to J.M.R.); and Spanish Ministerio de Ciencia e Innovación fellowships FPU (17/05866), FPI (BES-2016-077649), and Sara Borrell (CD18/00028), to A.d.l.F., M.J.R.-R., and M.T., respectively.

## Author contributions

M.R.C. and J.M.R. conceived the study; M.T., A.d.l.F.-A., M.R.C., and J.M.R. designed the study and analyzed the data; A.d.l.F.-A. and M.T. performed most of the experiments, with contributions from S.M.-M., A.A., M.J.R.-R., M.J.M.-O., and D.L.-M.; A.A., E.B.-K., I.G.-V., performed the proteomics and bioinformatics analysis under the supervision of J.V., who also provided ideas for the project; C.E.M., E.G.-I., S.M., A.F., L.M.-M., J.D.B., J.F.N., G.T.-T., and A.E. provided human tissue samples and ideas for the project; M.R.C. and J.M.R. wrote the manuscript with contributions from A.d.l.F.-A., M.T., E.B.-K., A.A., and J.V. All authors read and approved the manuscript.

## Competing interests

The authors declare no competing interests.
