## [Peer Review File · Nature Communications]

REVIEWER COMMENTS

Reviewer #1 (Remarks to the Author):

This ms describes research on the possible cause of syndromic (Marfan syndrome, MFS) and nonsyndromic, familial thoracic aortic aneurysm and dissection (TAAD). MFS is caused by a mutation in the fibrillin-1 gene and causes a severe dysregulation of the connective tissue homeostasis. Most prominent is this disease where it causes TAAD during the human lifetime. However, TAAD can be associated with mutation in several genes. Guo et al reported in 2013 (Am J Human Gen 93: 398) that a gain of function mutation in the PKG-1 gene is associated with TAAD in a few families. The effect of this mutation of PKG-1 (Arg177Gln) was later investigated in a mouse model - heterozygous for this mutation - by Schwaerzer et al (Nature Commun (2019)10:3533). These authors concluded that the activated PKG-1 increased the expression of Nox-4 and the production of H2O2 causing oxidative stress. The mechanism activated by PKG-1 and leading to increased Nox-4 expression remained unclear. The group of Redondo reported in 2017 (Oller et al Nature Med 23:200) that the NO producing enzyme Nos-2 is elevated in syndromic (MFS) and non-syndromic (Adams 1 deficiency) TAAD causing an increased NO production. In all the cited papers it was shown that TAAD occurred in respective mouse model. The present ms by Redondo reports now experiments which pathway is activated by increased long-term NO concentrations. As expected from our knowledge on the targets of NO, they find an increase in protein nitration and in the activity of soluble guanylyl cyclase and PKG-1. These features are associated with the typical de-arrangement of aortic connective tissue including TAAD. Silencing of PKG-1 in mice prevents NO induced TAAD. The generality for all TAAD syndromes remains to be established. However, this is an instructive piece of research and well done. The finding that PKG-1 is a major effector in some TAAD syndromes had been anticipated from the previous publications. Therefore, the novelty of the findings is very limited; especially because no experiments are presented to learn in which way long term activated PKG-1 induces TAAD. The clinical implication is unclear. The cautionary tale that long term elevation of cGMP by inhibition of PDE 5 or activation of soluble guanylyl cyclase may induce TAAD in humans was raised already by Schwaerzer et al (see above). This side effect of pharmacological manipulation of the cGMP levels has not been observed so far.

Minor point

Improve in line 866: ***p<0.0?

Reviewer #2 (Remarks to the Author):

Review Nat Comm; Aortic disease in Marfan syndrome is caused by overactivation of sGC-PRKG signaling by NO.

In general this is a very elegant story, building on their more recent knowledge that NO plays a key role in aortic disease and MFS in particular. The data provided here in MFS cells, mice and patients finally point into a direction of a treatable mechanism. The regression of AA is impressive, and more studies should be performed to focus on aortic repair rather than prevention of AA. Still I have a few questions remaining.

Results;

- VASP-S239 phosphorylation is known to be involved in F-actin accumulation (doi: 10.4161/cam.27351), which is a typical characteristic of MFS SMCs. Could F-actin staining be performed to show normalization of F-actin upon PRKG inhibition in MFS cells, and induction be observed in PRKG activating conditions in WT cells? MFS SMCs also have enhanced SMC markers (examples; ACTA2, calponin) mRNA / protein expression. Is this also normalized upon PRKG inhibition in MFS cells, and induction be observed in PRKG activating conditions in WT cells? (examples of MFS SMC phenotype studies: Arterioscler Thromb Vasc Biol. 2015 Apr; 35(4):960-72. doi: 10.1161/ATVBAHA.114.304412. AND Nat Genet. 2017 Jan; 49(1):97-109. doi: 10.1038/ng.3723.)
- Similarly, is fibrillin-1 fiber formation in 3 week SMC cultures rescued by PRKG inhibition in MFS cells, and disrupted by PRKG activation in WT cells? These are important experiments to show the

mechanism of aortic repair. Is it by normalizing the SMC phenotype and thus promoting normal SMC fibrillin-1 fiber formation and function?

Discussion; "...contractility via myosin light chain phosphatase activation and decreased calcium influx in VSMCs 37, 64. " Would this also explain the enhanced pathology observed in MFS and other AA patients when using calcium channel blockers? With data adding up, perhaps it is time to discuss if guidelines should be adjusted to prevent calcium channel blocker use as alternative BP regulator in patients at risk for development for AA? Ref: Elife 2015 Oct 27;4:e08648. doi: 10.7554/eLife.08648.

Minor: It seems a ref is missing when referring to aortic dissection at a normal aortic diameter. This is especially the case for type B dissections in Marfan. Ref: J Am Coll Cardiol. 2015 Jan 27;65(3):246-54. doi: 10.1016/j.jacc.2014.10.050.

Reviewer #3 (Remarks to the Author):

- Overall, this paper describes a well-controlled study supporting the role of the NO axis in Marfan syndrome and adding some new mechanistic data. The mouse work is strengthened by the correlation with clinical findings of increased circulating NO activity and variation in ambulatory venous pressures with episodes of hypotension in Marfan patients (Hillebrand et al 2016)
- The paper relies heavily on imaging and quantification of the imaging. This can, of course, be challenging because of the variability of antibody/epitope affinity, some level of nonspecific staining (that can be difficult to adjust for across antibodies) and selection bias in choosing sections for analysis. Were there any other protein studies such as Westerns to support the image quantification? The authors are careful to explain the process of elastin staining and quantification, but are less explicit on immunostaining. How many sections were used per sample? How were sections selected or importantly, excluded. Is analyses of those sections sufficient to represent quantification of the whole?
- The value of the Alcian stain (Figure 2f) and importance of information provided should be clarified.
- L-NAME, ODQ, and KT5823 almost completely blocked phosphorylation of VASP in WT SMCs, but not in MFS SMCs. Is this related to the doses of the inhibitors or are there other mechanisms that also lead to increase in P-VASP-S239 in MFS? Is this differential effect seen in vivo?
- Questions regarding lines 934-935 and figure 5d: when describing the nitro plasma index data in Figure 5d, the authors note that the whiskers on their distribution boxes extend from the minimum to maximum values but this does not seem to be the case. Does this represent outlier data that they didn't include in the statistics?
- The results showed that ODQ and KT5823 treatment lead to regression of aortic enlargement in MFS. Regression of an existing aneurysm is a clinically important endpoint so understanding the mechanism for this is critical. Is this regression associated with reversal of the Marfan associated matrix changes? Is there a decrease in wall thickness and GAG accumulation, repair of elastin breaks, restoration of more normal lamellar structure?
- Based on IF data (Figure 7d), the PRKG1 protein levels in MFS mice were similar to WT mice, indicating only PRKG1 activity was increased in MFS. Were cGC or Nos2 levels increased in MFS?
- A few typographical comments:
 - o Figure 6f: "WT" label is missing from the key
 - o Line 532: "u" should be changed to "μ" in "4 ul plasma".
 - o Figure 5a & b labeling is not clear

Reviewer #4 (Remarks to the Author):

Reviewer report

Manuscript ID: NCOMMS-20-20347-T

Title: Aortic disease in Marfan syndrome is caused by overactivation of sGC-PRKG signaling by NO

Comments

The authors aimed to investigate the roles and mechanism of NO-sGC-PRKG signaling in aortic disease in Marfan syndrome, and explore the potential of sGC and/or PKRG inhibition in treatment of aortic aneurysm and dissection in Marfan syndrome patients, in the different levels of cell, animal, and human models. This is an interesting topic and work. It is a well-organized and well-written manuscript. However, some drawbacks remain in current manuscript.

Major concerns:

The authors expect to confirm that nitric oxide (NO)-mediated sGC-PRKG signaling activation played important roles in aortic aneurysm and dissection in Marfan syndrome patients. NO is synthesized by nitric oxide synthase (NOS), including nNOS, iNOS, and eNOS. In inflammatory and pathogenesis conditions, iNOS-derived NO is significantly increased, and quickly reacts with superoxide anion to generate more toxic ONOO-, which is a pre-agent to cause protein nitration. Protein nitration is an important ROS/RNS-mediated protein post-translational modification (PTM), which occurs in amino acid residues Tyr and Trp. However, endogenous protein nitration is a low-abundance event - its occurrence rate is about 1 in 10⁶ tyrosine residues. Also, protein nitration can be reverse by endogenous denitrase. Thus protein nitration is not only a consequence of ROS/RNS damage but also protein nitration can change the protein function to involve in cell signaling. Based on this background, there are several major drawbacks.

1. Are there any proteins nitrated in the NO-sGC-PRKG signaling pathway? The author should detect the level of nitration of key molecules in the NO-sGC-PRKG signaling pathway. If yes, then the nitration site and its level in those proteins should be determined. In addition to plasma nitroproteomics, if nitroproteomics was also performed in the cell model, animal model, or human aortic aneurysm tissues of Marfan syndrome patients, it would be much better, and the authors have these cells and tissue samples. If any nitrated proteins and sites are identified in these cell and tissue samples, it would help much to explain the role and mechanism of NO-mediated sGC-PRKG signaling in aortic aneurysm and dissection in Marfan syndrome, even might find more meaningful results.
2. For plasma proteomics and protein nitration, because protein nitration is a low-abundance event, a preferential enrichment of tryptic nitropeptides is needed before LC-MS/MS analysis. However, after I carefully read online methods, no any enrichment strategy is used, why?
3. For the obtained plasma nitroproteins and tryptic nitropeptides in mouse (Extended data Table 1) and human (Extend data Table 2), which were identified with TMT-based quantitative proteomics, some important information is missing in these two tables, including protein accession ID, modified site in a protein amino acid sequence, ion score to evaluate the quality of each MS/MS spectrum, nitration abundance, ratio of nitration in disease vs. control, and ammonium ion at m/z 181.06 for mononitrated tyrosine (detected, or not detected) that is a characteristic ion to confirm the existence of nitro (-NO₂) group in a amino acid sequence.
4. Mass spectrometry identification of nitropeptides are very challenging, therefore, all MS/MS spectra of 50 nitropeptides from mouse plasma and 41 nitropeptides from human plasma should be collected in extended data figures 6 and 7. There are some nitropeptides with a very long amino acid sequence, such as serotransferrin in Extended Data Table 1, its corresponding nitropeptide includes 48 amino acid residues, how about the quality of its MS/MS spectrum? Also, overview of all identified nitropeptides in Extended Data Table 1, many amino acid residue "I" (isoleucine) are contained in those nitropeptides, the mass of leucine (L) and isoleucine (I) is very close, it is also necessary to present all MS/MS spectra.
5. For Figure 5D, why did you select those 7 tryptic nitropeptides from Extended Data Table 2 to calculate the nitrated plasma index (NPI)? Is it reasonable? No reason is given. Also, these tryptic nitropeptides did not exist in human plasma, but nitroproteins exist in plasma. How did you to use this NPI parameter in real clinical practice.
6. Did you find any non-nitrated peptides for those identified nitroproteins? If so, it should be listed in the Extended Data table.
7. Is there any difference between nitro-Tyr and nitro-Trp?
8. Is there any difference in plasma protein nitration between mouse and human?
9. How about the overall protein nitration levels in the plasma, cell and tissues, which can be tested by Western blot and/or immunohistochemistry?

Minor concerns:

1. Line 192: how is the nitrated plasma index (NPI) calculated? Did you test its reproducibility of NPI in your samples?
2. Line 262: For "data not shown", can you collect them as an Extended Data material?
3. Lines 321-324: a large clinical sample size should be used to test plasma cGMP and protein nitration biomarkers.
4. Lines 532: "4 ul plasma" should be "4 μ l plasma".
5. Line 604: "as the" should be "as the".
6. Line 527: nitration is a low abundance event, why are there no enrichment of nitropeptides before LC-MS/MS analysis?

REVIEWER COMMENTS

Reviewer #1 (Remarks to the Author):

This ms describes research on the possible cause of syndromic (Marfan syndrome, MFS) and nonsyndromic, familial thoracic aortic aneurysm and dissection (TAAD). MFS is caused by a mutation in the fibrillin-1 gene and causes a severe dysregulation of the connective tissue homeostasis. Most prominent is this disease where it causes TAAD during the human lifetime. However, TAAD can be associated with mutation in several genes. Guo et al reported in 2013 (Am J Human Gen 93: 398) that a gain of function mutation in the PKG-1 gene is associated with TAAD in a few families. The effect of this mutation of PKG-1 (Arg177Gln) was later investigated in a mouse model - heterozygous for this mutation - by Schwaerzer et al (Nature Commun (2019)10:3533). These authors concluded that the activated PKG-1 increased the expression of Nox-4 and the production of H₂O₂ causing oxidative stress. The mechanism activated by PKG-1 and leading to increased Nox-4 expression remained unclear. The group of Redondo reported in 2017 (Oller et al Nature Med 23:200) that the NO producing enzyme Nos-2 is elevated in syndromic (MFS) and non-syndromic (Adamts 1 deficiency) TAAD causing an increased NO production. In all the cited papers it was shown that TAAD occurred in respective mouse model. The present ms by Redondo reports now experiments which pathway is activated by increased long-term NO concentrations. As expected from our knowledge on the targets of NO, they find an increase in protein nitration and in the activity of soluble guanylyl cyclase and PKG-1. These features are associated with the typical de-arrangement of aortic connective tissue including TAAD. Silencing of PKG-1 in mice prevents NO induced TAAD. The generality for all TAAD syndromes remains to be established. However, this is an instructive piece of research and well done.

The finding that PKG-1 is a major effector in some TAAD syndromes had been anticipated from the previous publications. Therefore, the novelty of the findings is very limited; especially because no experiments are presented to learn in which way long term activated PKG-1 induces TAAD. The clinical implication is unclear. The cautionary tale that long term elevation of cGMP by inhibition of PDE 5 or activation of soluble guanylyl cyclase may induce TAAD in humans was raised already by Schwaerzer et al (see above). This side effect of pharmacological manipulation of the cGMP levels has not been observed so far.

We thank the reviewer for the positive comments on the instructive nature of our research and for indicating that the work was well done.

We agree that establishing the role of NO signaling pathways in syndromic and non-syndromic TAADs is of the utmost importance. In fact, this is a current and major line of research in our laboratory. However, we consider that these studies are beyond the scope of this contribution, as a few additional years will be required to complete them properly, and we hope that they will be the subject of a contribution focused on this topic.

We also agree with the reviewer on the relevance of elucidating the mechanisms by which the long-term activation of PKG induces TAAD. In the revised version of the manuscript, we have incorporated data showing that dysregulation of actin cytoskeleton dynamics is a central mechanism underlying the induction of TAAD by long term

activation of the NO-PKG pathway. In this regard, we now show a sharp decrease in filamentous actin (F-actin) accumulation in WT VSMCs treated with DetaNO or 8-Br-cGMP, and in untreated MFS cells (**new Figure 2a-2b**). Conversely, pharmacological inhibition of PKG in MFS cells restored F-actin formation to levels of control cells (**new Figure 2b**), supporting the notion that the NO-PKG pathway is essential for actin cytoskeleton dynamics regulation and therefore for effective cell contraction in MFS. In this regard, we also show that PKG activity regulates the expression of contractile proteins in VSMCs, including alpha-smooth muscle actin (*Acta2*), smooth muscle protein 22 alpha (*Tagln2*), and calponin-1 (*Cnn1*). RT-qPCR experiments showed substantial increases in *Acta2*, *Cnn1* and *Tagln2* mRNA levels in WT VSMCs treated with DetaNO or 8-Br-cGMP (**new Figure 2c**) and also in MFS VSMCs (**new Figure 2d**). Further supporting a major role of PKG as regulator of actin cytoskeleton dynamics and expression of VSMC contractile markers, we show that pharmacological inhibition of PKG and *Prkg1* silencing decreased the expression of these markers in MFS cells to normal levels (**new Figure 2d** and **new Supplementary Figure 12a**) and increased actin fiber formation to normal levels (new Figure 2b and **new Supplementary Figure 12b**), in agreement with the results obtained upon pharmacological *Prkg* inhibition in VSMCs.

Since contractile dysfunction of VSMC leads to altered aortic structure and TAAD, these results would mechanistically link the long-term activation of PRKG-1 to TAAD.

We hope that these results satisfy the reviewer's concern about the novelty. We would like to emphasize that, regardless of this new set of data, the original version of the manuscript already contained important novel findings, including that: i) NO donors induce MFS-like aortopathy in WT mice; ii) sGC and PRKG1 are over-activated in MFS mice and patients and in WT mice treated with NO donors iii) the identification of potential biomarkers for the follow-up of patients with MFS; iv) the identification of sGC and PRKG1 as potential targets for therapeutic intervention in MFS aortopathy. We believe that these results are not only clearly novel, but also of considerable translational potential.

Minor point

Improve in line 866: “**p<0.0?

We have corrected the typographical mistake corresponding to line 866 in the original manuscript. Thank you.

Reviewer #2 (Remarks to the Author):

Review Nat Comm; Aortic disease in Marfan syndrome is caused by overactivation of sGC-PRKG signaling by NO.

In general this is a very elegant story, building on their more recent knowledge that NO plays a key role in aortic disease and MFS in particular. The data provided here in MFS cells, mice and patients finally point into a direction of a treatable mechanism. The regression of AA is impressive, and more studies should be performed to focus on aortic repair rather than prevention of AA. Still I have a few questions remaining.

We thank the Reviewer for his/her careful reading of the manuscript and for raising important issues that we have taken into account to improve the revised version. We greatly appreciate the comments on the elegance of the story and the regression of AA.

Results;

- VASP-S239 phosphorylation is known to be involved in F-actin accumulation (doi: 10.4161/cam.27351), which is a typical characteristic of MFS SMCs. Could F-actin staining be performed to show normalization of F-actin upon PRKG inhibition in MFS cells, and induction be observed in PRKG activating conditions in WT cells? MFS SMCs also have enhanced SMC markers (examples; ACTA2, calponin) mRNA / protein expression. Is this also normalized upon PRKG inhibition in MFS cells, and induction be observed in PRKG activating conditions in WT cells? (examples of MFS SMC phenotype studies: *Arterioscler Thromb Vasc Biol.* 2015 Apr;35(4):960-72. doi: 10.1161/ATVBAHA.114.304412. AND *Nat Genet.* 2017 Jan;49(1):97-109. doi: 10.1038/ng.3723.)
- Similarly, is fibrillin-1 fiber formation in 3 week SMC cultures rescued by PRKG inhibition in MFS cells, and disrupted by PRKG activation in WT cells? These are important experiments to show the mechanism of aortic repair. Is it by normalizing the SMC phenotype and thus promoting normal SMC fibrillin-1 fiber formation and function?

The Reviewer raises an important issue and we have made a major effort to investigate the mechanism of aortic repair. As the Reviewer points out, VASP-S239 phosphorylation is indeed known to impair filamentous actin (F-actin) formation (Benz, P.M., et al., 2009, *J Cell Sci*, PMID 19825941; Doppler, H., et al., 2013, *Cell Adh Migr*, PMID 24401601). Following his/her suggestion, we stained F-actin as a complementary read-out of VASP-S239 differential phosphorylation. Our results show a marked decrease in F-actin accumulation in WT cells treated with DetaNO or 8-Br-cGMP and in untreated VSMCs from *Fbn1*^{C1039G/+} mice (**New Figures 2a-2b**). Consequently, PRKG inhibition by KT5823 in *Fbn1*^{C1039G/+} VSMCs restored F-actin accumulation to normal levels (**New Figure 2b**). Similarly, *Prkg1* knockdown in these cells also markedly increased F-actin formation (**New Supplementary Figure 12b**).

As also suggested by the Reviewer, we have assessed the expression levels of contractility markers, including α -smooth muscle actin (Acta2), smooth muscle protein 22 alpha (Tagln2), and calponin-1 (Cnn1). RT-qPCR analysis showed a substantial

increase in *Acta2*, *Tagln2*, and *Cnn1* mRNA expression in WT VSMCs treated with Deta-NO or 8-Br-cGMP and, as reported, in MFS VSMCs (**New Figure 2c**). Notably, the expression of these markers regressed to normal levels in MFS cells upon pharmacological inhibition of PRKG (**New Figure 2d**) or following its silencing (**New Supplementary Figure 12a**).

These data are described (pages 5-6) and discussed (pages 14-15) in the revised version.

We have also followed the Reviewer's suggestion to investigate the role of the NO signaling pathway in regulating fibrillin-1 fiber deposition. Work by other investigators showed that extracellular fibrillin-1 deposition was irregular and less abundant in smooth muscle cells derived from human induced pluripotent stem cells harboring the *Fbn1*^{C1242Y} pathogenic variant than in control cells (Granata, A., et al., Nat Genet 2017; PMID 27893734). However, we found that fibrillin-1 was barely detectable in VSMCs derived from *Fbn1*^{C1039G/+} mice, and *Prkg1* silencing in these cells did not substantially increase its levels (**New Supplementary Figure 12c**). Moreover, activation of the NO signaling pathway in WT VSMCs upon treatment with 8-Br-cGMP did not substantially modify fibrillin-1 fiber formation (**New Supplementary Figure 1**). These results, described in page 6 (paragraph 1) and page 10 (paragraph 2), suggest that fibrillin-1 secretion and its capacity to form fibers might be differentially compromised in distinct pathogenic variants, regardless the degree of activation of the NO-sGC-PRKG pathway. Given that the NO signaling pathway regulates the phenotype of VSMCs and that distinct *Fbn1* pathogenic variants might differ in their capacity to form extracellular fibers, regardless of NO signaling activation, we believe that our results support the notion that MFS aortopathy involves a critical contribution from dysregulation of actomyosin cytoskeleton dynamics in MFS VSMCs by overactivation of the NO-sGC-PRKG pathway.

Discussion; "...contractility via myosin light chain phosphatase activation and decreased calcium influx in VSMCs 37, 64. " Would this also explain the enhanced pathology observed in MFS and other AA patients when using calcium channel blockers? With data adding up, perhaps it is time to discuss if guidelines should be adjusted to prevent calcium channel blocker use as alternative BP regulator in patients at risk for development for AA? Ref: Elife 2015 Oct 27;4:e08648. doi: 10.7554/eLife.08648.

We thank the reviewer for this suggestion. We have accordingly modified the Discussion of the revised version of the manuscript (page 14, last paragraph).

Minor: It seems a ref is missing when referring to aortic dissection at a normal aortic diameter. This is especially the case for type B dissections in Marfan. Ref: J Am Coll Cardiol. 2015 Jan 27;65(3):246-54. doi: 10.1016/j.jacc.2014.10.050.

We thank the reviewer for this observation and we have included this reference in the new version of the manuscript (page 13, last paragraph)

Reviewer #3 (Remarks to the Author):

- Overall, this paper describes a well-controlled study supporting the role of the NO axis in Marfan syndrome and adding some new mechanistic data. The mouse work is strengthened by the correlation with clinical findings of increased circulating NO activity and variation in ambulatory venous pressures with episodes of hypotension in Marfan patients (Hillebrand et al 2016)
- The paper relies heavily on imaging and quantification of the imaging. This can, of course, be challenging because of the variability of antibody/epitope affinity, some level of nonspecific staining (that can be difficult to adjust for across antibodies) and selection bias in choosing sections for analysis. Were there any other protein studies such as Westerns to support the image quantification? The authors are careful to explain the process of elastin staining and quantification, but are less explicit on immunostaining. How many sections were used per sample? How were sections selected or importantly, excluded. Is analyses of those sections sufficient to represent quantification of the whole?

We thank the Reviewer for his/her comments that our study is well controlled and that it adds new mechanistic data.

We agree with the Reviewer on the importance of appropriate quantification of the imaging data, and this comment alerted us to the fact that in the original manuscript we neglected to properly explain the details of the image selection and quantification. We mostly analyzed protein expression using immunostaining techniques because this allowed us to visualize patterns of expression in different cell types and aortic locations; moreover, these techniques are more suitable than immunoblot analysis given the very limited amounts of proteins that can be obtained from mouse aortic tissue. We now include experimental details of cell and tissue immunostaining quantification in the Methods section (last paragraph of page 17 (cells) and page 20 (aortic tissue)). In addition, we have increased the number of images used for quantification of each experimental condition in experiments included in the revised Figure 1. We believe that the description and quantification of these experiments is now sufficiently clear, and we apologize for not having explained this properly in the original manuscript.

- The value of the Alcian stain (Figure 2f) and importance of information provided should be clarified.

We have followed the Reviewer's recommendation by expanding the information provided on the importance of proteoglycans in aortic structure and function, and we have also included three new references in the revised manuscript. We now indicate that "Proteoglycans play essential roles in preserving aortic structure and function^{10, 11, 12} by regulating elastic fiber assembly and smooth muscle cell proliferation" (page 3, paragraph 1). In addition, we have modified the Results text about Alcian staining to clarify that PG accumulation and elastic fiber fragmentation are central features of medial degeneration and that they were analyzed by Alcian Blue and VGE staining, respectively, in **new Figures 3f and 3g** (page 7, first paragraph)

- L-NAME, ODQ, and KT5823 almost completely blocked phosphorylation of VASP in WT SMCs, but not in MFS SMCs. Is this related to the doses of the inhibitors or are

there other mechanisms that also lead to increase in P-VASP-S239 in MFS? Is this differential effect seen *in vivo*?

The Reviewer is right that the NO-sGC-PRKG pathway inhibitors used more efficiently block VASP phosphorylation in WT than in MFS SMCs. Although we cannot exclude PRKG activation in MFS cells via mechanisms unrelated to this pathway, we believe that this differential effect is related to the short duration of treatment with the inhibitors (1 hour). This short treatment may not be sufficient to reverse the high baseline levels of pVASP observed in SMCs from MFS mice. In the *in vivo* models, we observed that the levels of pVASP in MFS mice reverted to those of WT when the mice were treated with ODQ for 21 days. Similarly, we found a marked reversion of pVASP levels in MFS mice treated with KT5823 for 7 days. We now hypothesize that this partial reversion with KT5823 is due to the short treatment duration (page 10, paragraph 1). Supporting this, we show that *Prkg1* silencing leads to near complete reversion of pVASP levels after 28 days (**new Figures 8e-8f**). Please note that **new Figures 8d-8k** now include data from a larger number of mice per group than in the previous version, as we had to include more mice to address another of the Reviewer's concerns (see below).

- Questions regarding lines 934-935 and figure 5d: when describing the nitro plasma index data in Figure 5d, the authors note that the whiskers on their distribution boxes extend from the minimum to maximum values but this does not seem to be the case. Does this represent outlier data that they didn't include in the statistics?

We agree with the Reviewer and have corrected the legend to **new Figure 6d** to indicate that whiskers extend 1.5 times above and below the interquartile range. Values outside this range are usually considered outliers and are typically excluded from statistical analysis, thus improving data significance. However, we believed it was more appropriate to include them in this particular case because the cohort size was relatively small. All data were therefore included in the statistical analysis of the nitro-proteomics study.

- The results showed that ODQ and KT5823 treatment lead to regression of aortic enlargement in MFS. Regression of an existing aneurysm is a clinically important endpoint so understanding the mechanism for this is critical. Is this regression associated with reversal of the Marfan associated matrix changes? Is there a decrease in wall thickness and GAG accumulation, repair of elastin breaks, restoration of more normal lamellar structure?

The Reviewer raises an important issue and we have analyzed the effects of ODQ and KT5823 on the restoration of the vascular wall in MFS mice. We performed a histological analysis of AaAo cross-sections and found that ODQ significantly restores elastic-fiber fragmentation and reduces aortic wall thickness in MFS mice after 21 days of treatment (**new Supplementary Figure 11a**). However, we did not observe these effects after 7 days of treatment of these mice with KT5823 (**new Supplementary Figure 11b**). As with the reversion of pVASP levels in MFS mice, this short-term treatment with KT5823 may be too short to restore the vascular wall structure of MFS mice. Again, this possibility is supported by our experiments showing that after 28 days *Prkg1* silencing does lead to a near complete reversal of aortic wall thickening and elastic-fiber fragmentation in MFS mice (new Figures 8j-8k).

- Based on IF data (Figure 7d), the PRKG1 protein levels in MFS mice were similar to WT mice, indicating only PRKG1 activity was increased in MFS. Were cGC or Nos2 levels increased in MFS?

We now present data showing that mRNA and protein expression levels of the sGC alpha and beta subunits are similar in the aortas of MFS and WT mice (**new Supplementary Figures 6a-6b**). Furthermore, consistent with the IF data from Figure 7d in the original manuscript, the Prkg1 protein and RNA expression levels in the aorta are also similar in both genotypes (**new Supplementary Figures 6c-6d**). These new data are described on page 7, last paragraph. Regarding Nos2, we have previously shown that its mRNA and protein levels are markedly increased in aortas of MFS mice relative to littermate controls and also in aortic sections of MFS patients (Reference 19 of the manuscript). Together, these results indicate that the activation of the NO-sGC-PRKG pathway can be regulated in the aorta at both the transcriptional and post-transcriptional level.

- A few typographical comments:
 - o Figure 6f: “WT” label is missing from the key
 - o Line 532: “u” should be changed to “ μ ” in “4 ul plasma”.
 - o Figure 5a & b labeling is not clear

We have corrected the typographical errors in Figures 6f and line 532. In addition, we have extended the legend to **new Figures 6a and 6b** to clarify its labelling. Thank you.

Reviewer #4 (Remarks to the Author):

Reviewer report

Manuscript ID: NCOMMS-20-20347-T

Title: Aortic disease in Marfan syndrome is caused by overactivation of sGC-PRKG signaling by NO

Comments

The authors aimed to investigate the roles and mechanism of NO-sGC-PRKG signaling in aortic disease in Marfan syndrome, and explore the potential of sGC and/or PKRG inhibition in treatment of aortic aneurysm and dissection in Marfan syndrome patients, in the different levels of cell, animal, and human models. This is an interesting topic and work. It is a well-organized and well-written manuscript. However, some drawbacks remain in current manuscript.

We thank the Reviewer for his/her comments on the interest of the study, and also for raising important issues that we have taken into account to improve the manuscript. We have performed further experiments that support and validate our hypothesis and address the Reviewer's concerns.

Major concerns:

The authors expect to confirm that nitric oxide (NO)-mediated sGC-PRKG signaling activation played important roles in aortic aneurysm and dissection in Marfan syndrome patients. NO is synthesized by nitric oxide synthase (NOS), including nNOS, iNOS, and eNOS. In inflammatory and pathogenesis conditions, iNOS-derived NO is significantly increased, and quickly reacts with superoxide anion to generate more toxic ONOO-, which is a pre-agent to cause protein nitration. Protein nitration is an important ROS/RNS-mediated protein post-translational modification (PTM), which occurs in amino acid residues Tyr and Trp. However, endogenous protein nitration is a low-abundance event - its occurrence rate is about 1 in 10⁶ tyrosine residues. Also, protein nitration can be reverse by endogenous denitrase. Thus protein nitration is not only a consequence of ROS/RNS damage but also protein nitration can change the protein function to involve in cell signaling. Based on this background, there are several major drawbacks.

1. Are there any proteins nitrated in the NO-sGC-PRKG signaling pathway? The author should detect the level of nitration of key molecules in the NO-sGC-PRKG signaling pathway. If yes, then the nitration site and its level in those proteins should be determined. In addition to plasma nitroproteomics, if nitroproteomics was also performed in the cell model, animal model, or human aortic aneurysm tissues of Marfan syndrome patients, it would be much better, and the authors have these cells and tissue samples. If any nitrated proteins and sites are identified in these cell and tissue samples, it would help much to explain the role and mechanism of NO-mediated sGC-PRKG signaling in aortic aneurysm and dissection in Marfan syndrome, even might find more meaningful results.

This is a very important issue, and we have made a major effort to investigate the mechanism of NO-mediated signaling in aortic aneurysm and dissection in Marfan syndrome. NOS enzymes, NO, sGC, and PRKG constitute the core of the NO-sGC-PRKG signaling pathway, but we expect that the pathway also includes numerous proteins upstream of NOS and downstream of PRKG, many of them as yet unidentified. None of the core pathway components were detected in the proteomics analysis of plasma samples. We therefore followed the Reviewer's suggestion to perform a high-throughput quantitative proteomics analysis comparing the nitro-proteome profile of MFS and WT aortic tissue. This analysis showed a significant increase in protein nitration levels in MFS aorta and a consistent up-regulation of 24 aortic nitro-proteins (**new Figure 6e, new Supplementary Figure 8, and new Supplementary Data 5**). These data further support our hypothesis that the NO-sGC-PRKG signaling pathway is activated in Marfan syndrome.

Although the nitration levels of core components were similar in WT and MFS aorta, we are cautious about making statements that exclude the possible differential nitration of these proteins because some of them are of low abundance, nitration is a low-abundance event (as commented by the reviewer), and the method we used to characterize protein nitration only detects the most abundant nitration events. Nevertheless, in MFS we detected a marked increase in nitrated Acta2, an abundant protein in the aorta (**new Figure 6f**). As discussed in the revised manuscript, Tyr-actin nitration impairs actin cytoskeleton dynamics. Accordingly, we now show a substantial decrease of actin filaments in MFS cells and in WT cells treated with PRKG activators (**new Figure 2**). These results therefore suggest that increased Acta2 nitration in MFS might impair actin filament formation. Indeed, the NO-sGC-PRKG pathway has been linked to the tuning of cell contractility, a feature that involves the indirect connection of the actomyosin cytoskeleton with the extracellular matrix. In this regard, we have detected a substantial increase of the nitration of 7 additional cytoskeletal and extracellular matrix proteins (**new Fig. 6f**), that might also play a role in contractility regulation.

2. For plasma proteomics and protein nitration, because protein nitration is a low-abundance event, a preferential enrichment of tryptic nitropeptides is needed before LC-MS/MS analysis. However, after I carefully read online methods, no any enrichment strategy is used, why?

We originally considered the possibility of applying enrichment methods before the LC-MS/MS analysis. The major approaches to enrich nitrated species are chemical derivatization and affinity-based methods, but each of these approaches has some disadvantages regarding specificity, yield and/or unambiguous identification of the exact nitration sites. We decided to perform a high-throughput proteomics analysis to determine protein nitration levels because this approach is unbiased, does not introduce chemical modifications, is based on stable-isotope labelling-quantification (the method of choice for chemical quantification), has the required depth to identify a large enough number of nitration sites in our hands, and allows the precise and unambiguous identification of nitrated sites from the MS/MS spectra.

Nevertheless, we explored the possibility of using specific α -Nitro Tyrosine antibodies as tools for assessing protein nitration in plasma from patients and healthy individuals. However, the experiments with these antibodies were not sufficiently reproducible in our hands to support conclusions. We first tested them in immunoblot analysis of plasma from untreated WT and MFS mice and in plasma from WT mice treated with the ISMN NO donor. In the initial experiment, there seemed to be a significant upregulation in nitro-levels of some proteins in MFS and in WT mice treated with ISMN for 7 or 28 days (panel A of Figure 1 for the reviewer; see below). However, we could not reproduce these results when plasma samples from additional mice were used (**panels B and C of Figure 1 for the reviewer**). The lack of reproducibility of these results was confirmed with samples from additional mice in further experiments.

Figure 1 for the reviewer. Nitro-Tyr immunoblot analysis of plasma proteins in mouse samples. (A) Nitro-Tyr immunoblot analysis of plasma samples from 3 mice per group of untreated WT and MFS mice and WT mice treated with ISMN for 2 days (2d), 7 days (7d), or 28 days (28d). (B,C) Nitro-Tyr immunoblot analysis of plasma samples from 3 mice per group of untreated WT and MFS mice and WT mice treated with ISMN for 7 days. The position of molecular weight markers is indicated

3. For the obtained plasma nitroproteins and tryptic nitropeptides in mouse (Extended data Table 1) and human (Extend data Table 2), which were identified with TMT-based quantitative proteomics, some important information is missing in these two tables, including protein accession ID, modified site in a protein amino acid sequence, ion

score to evaluate the quality of each MS/MS spectrum, nitration abundance (spectral count), ratio of nitration in disease vs. control, and immonium ion at m/z 181.06 for mononitrated tyrosine (detected, or not detected) that is a characteristic ion to confirm the existence of nitro (-NO₂) group in a amino acid sequence.

The point is well taken. The additional information requested by the reviewer has been added to the **new Supplementary data 1 and 3** files. We also provide an additional **new Supplementary data 5** file containing the same set of data for the nitration analysis in mouse ascending aorta samples.

4. Mass spectrometry identification of nitropeptides are very challenging, therefore, all MS/MS spectra of 50 nitropeptides from mouse plasma and 41 nitropeptides from human plasma should be collected in extended data figures 6 and 7. There are some nitropeptides with a very long amino acid sequence, such as serotransferrin in Extended Data Table 1, its corresponding nitropeptide includes 48 amino acid residues, how about the quality of its MS/MS spectrum?

Following the Reviewer's suggestion, the MS/MS spectra of all nitro-peptides identified in human and mouse plasma are now presented in **new Supplementary data 2 and 4** files. These spectra are also freely available at <http://www.peptideatlas.org/PASS/PASS01528>, as indicated in the Data availability section. The good quality of these spectra readily enabled assignation of MS/MS fragments to peptide sequences and unambiguous peptide identification.

Also, overview of all identified nitropeptides in Extended Data Table 1, many amino acid residue "I" (isoleucine) are contained in those nitropeptides, the mass of leucine (L) and isoleucine (I) is very close, it is also necessary to present all MS/MS spectra.

As the reviewer points out, leucine (L) and isoleucine (I) are indistinguishable in MS/MS spectra because they share the same incremental amino acid mass. For this reason, most search engines and decoy database generator software programs for large scale proteomics consider these amino acids as the same entity and use a common amino acid symbol. Thanks to the Reviewer's comment, we have noticed that the sequences listed in the original submission still contained the common amino acid symbol. This mistake has been corrected in the new Supplemental Data 1 and 3 files, indicating the correct peptide sequence of the identified protein. To check for errors in protein assignation, we have carefully revised and updated nitrated peptide sequences after performing an *in silico* bioinformatics analysis. We first checked information contained in the Swiss-Prot and TrEMBL databases via Uniprot knowledgebase (<https://www.uniprot.org/>) (already used in the workflow analysis). We also searched for the corresponding mRNA transcript of each protein in the NCBI Nucleotide database, programmatically accessing the ExPASy Translate tool (<https://web.expasy.org/translate/>) to obtain the corresponding peptide sequence for the nitrated proteins presented. Finally, we compared sequences with the manually annotated sequences on the Swiss-Prot database, to ensure that amino acid sequences are coherently described in all databases used.

5. For Figure 5D, why did you select those 7 tryptic nitropeptides from Extended Data Table 2 to calculate the nitrated plasma index (NPI)? Is it reasonable? No reason is given. Also, these tryptic nitropeptides did not exist in human plasma, but nitroproteins exist in plasma. How did you use this NPI parameter in real clinical practice.

The idea behind the nitrated peptide index is to select a panel of nitrated peptides that best measures the extent of nitration in plasma, in the same way that biomarker panels are constructed with sets of biomarker proteins. Indeed, there are many studies reporting the detection of PTM signatures as potential biomarkers in clinical mass spectrometry (Mnatsakanyan, R., et al., 2018, Expert Rev Proteomics 15:515-35; PMID: 29893147). We therefore selected the 7 nitrated peptides that yielded statistically significant changes in the comparisons of Marfan and control groups (all significantly changing peptides were more abundant in Marfan samples). Note that the NPI is not a measure of the global nitration level in plasma.

To make this point clearer, we have rewritten the sentence in the main text as follows: *“To provide a set of candidate nitrated peptides as biomarkers for clinical diagnosis or prognosis, we selected the nitrated peptides that were significantly upregulated in MFS patients (Fig. 6c). The quantitative values of these peptides were combined to obtain a nitrated plasma index (NPI), which provides a measure of the increase in nitration.”*

We also agree with the Reviewer in that, although in this study we used bottom-up proteomics (identifying proteins from their peptides), the entities that we actually detected in plasma are nitrated proteins. For this reason, and to improve biological interpretation of the data, we have condensed the nitration information at the protein level in **new Fig. 6a and 6b** to show not nitrated peptides, but nitrated proteins. The information at the individual peptide level, including the nitrated sites, is maintained in **new Supplementary Data files 1, 3, and 5**.

6. Did you find any non-nitrated peptides for those identified nitroproteins? If so, it should be listed in the Extended Data table.

We indeed identified the corresponding non-modified peptide for nearly all cases of nitropeptide assignment, reinforcing the accuracy of nitration site identification. This information has been included in **new supplementary Data 1, 3, and 5**. Moreover, all the nitrated proteins have been identified by several other non-modified peptides; this information has also been included in these tables.

We should note here that we used the non-modified peptides to quantify the proteins, and in our statistical model the quantification at the modified peptide level is corrected by the protein value, so that they are not affected by protein changes (Bagwan, N., et al., 2018, Cell Rep, 23:3685-97, PMID: 29925008).

7. Is there any difference between nitro-Tyr and nitro-Trp?

The nitration percentage of Tyr and Trp in mouse and human plasma and in mouse aorta is now indicated in the Results section (page 8, last paragraph; page 9, paragraphs 1 and 2). These results show that nitro-Tyr is markedly more abundant than nitro-Trp in mouse and human samples.

8. Is there any difference in plasma protein nitration between mouse and human?

We observed a similar positive shift in plasma nitration levels in MFS (**new Fig. 6a, 6b**). The nitro-proteins identified in both experiments correspond to the most abundant plasma proteins: albumin, fibrinogens, and immunoglobulins (**new Supplementary Data 1 and 3**). This suggests that nitration has a very similar behaviour in both kind of samples. These data are commented on in the Results section (page 9, paragraph 1).

9. How about the overall protein nitration levels in the plasma, cell and tissues, which can be tested by Western blot and/or immunohistochemistry?

As commented in our response to major concern #2, western blotting with an anti-Nitro-Tyr antibody detected up-regulated nitration levels in plasma from MFS and ISMN-treated mice. However, the technical reproducibility of this approach was too low to trust the results.

Minor concerns:

1. Line 192: how is the nitrated plasma index (NPI) calculated? Did you test its reproducibility of NPI in your samples?

The methodology used to calculate NPI is already explained in the Proteomics section of the Online Methods: “Plasma-protein nitration in each patient was measured as the nitrated plasma index (NPI), defined as the weighted mean of the 7 most upregulated nitro-peptides”. This is done by applying the WSSP model, which assigns a weight to each measurement that corresponds to the inverse of its variance, according to error propagation theory (Trevisan-Herraz, M., et al., 2019, *Bioinformatics* 35:1594-6; PMID 30252043).

NPI reproducibility was tested by measuring it separately in independent subcohorts of our study population. Although the samples came from three different hospitals, the significance of the difference between healthy donors and MFS patients is clearly reproduced in both subcohorts (see **Figure 2 for the Reviewer**, below). However, it will be necessary to conduct a larger longitudinal study in the future to confirm applicability of the NPI in clinical practice.

Figure 2 for the reviewer. Boxplots showing the sample distribution based on the Nitrated Plasma Index (NPI) between MFS patients and healthy donors divided by cohorts to assess the reproducibility of this estimation. The samples come from (A) a healthy population (n=16) and a MFS population (n=14) from a hospital in Barcelona (Cohort1); and (B) a healthy population (n=14) and a MFS population (n=9), from hospitals in Santander and Ghent (Cohort 2). A significant difference between conditions is confirmed in each cohort. Each data point denotes an individual, boxes represent the interquartile range (IQR), the line in the box shows the median value, and the whiskers extend 1.5 times above and below the IQR. * $p < 0.05$ by unpaired two-tailed Student's *t*-test.

2. Line 262: For “data not shown”, can you collect them as an Extended Data material?

The sentence that included “data not shown” has been removed from the Discussion

3. Lines 321-324: a large clinical sample size should be used to test plasma cGMP and protein nitration biomarkers.

We completely agree with Reviewer’s comment and we have modified the Discussion accordingly (page 13, last paragraph).

4. Lines 532: “4 ul plasma” should be “4 μ l plasma”.

Corrected.

5. Line 604: “as the” should be “as the”.

Corrected, adding “by the” instead of “as the”.

6. Line 527: nitration is a low abundance event, why are there no enrichment of nitropeptides before LC-MS/MS analysis?

This minor concern has been answered in our response to major concern #2, explaining the motivation for using an unbiased high-throughput proteomics analysis.

REVIEWERS' COMMENTS

Reviewer #1 (Remarks to the Author):

The paper contains now initial experiments that suggest that activated PKG I interferes with the accumulation of filamentous actin. Apparently long term activation of PKG I affects the cytoskeletal dynamics. I further appreciate that the authors scaled down the potential clinical implication of their results!

Reviewer #2 (Remarks to the Author):

I have nothing further, since the authors have implemented all my suggestions.

Reviewer #3 (Remarks to the Author):

The concerns and issues have been carefully addressed through revisions and additional experimentation.

Reviewer #4 (Remarks to the Author):

The revised manuscript is significantly improved. My concerns have been resolved. No more comments.